# CMG helicase disassembly is essential and driven by two pathways in budding yeast

Cristian Polo Rivera [1], Tom D Deegan [1,2]✉ & Karim P M Labib [1]✉

## Abstract

**The CMG helicase is the stable core of the eukaryotic replisome and is ubiquitylated and disassembled during DNA replication termination. Fungi and animals use different enzymes to ubiquitylate the Mcm7 subunit of CMG, suggesting that CMG ubiquitylation arose repeatedly during eukaryotic evolution. Until now, it was unclear whether cells also have ubiquitin-independent pathways for helicase disassembly and whether CMG disassembly is essential for cell viability. Using reconstituted assays with budding yeast CMG, we generated the *mcm7-10R* allele that compromises ubiquitylation by SCF[Dia2]. *mcm7-10R* delays helicase disassembly in vivo, driving genome instability in the next cell cycle. These data indicate that defective CMG ubiquitylation explains the major phenotypes of cells lacking Dia2. Notably, the viability of *mcm7-10R* and *dia2Δ* is dependent upon the related Rrm3 and Pif1 DNA helicases that have orthologues in all eukaryotes. We show that Rrm3 acts during S-phase to disassemble old CMG complexes from the previous cell cycle. These findings indicate that CMG disassembly is essential in yeast cells and suggest that Pif1-family helicases might have mediated CMG disassembly in ancestral eukaryotes.**

**Keywords** CMG Helicase; SCF[Dia2]; DNA Replication; Ubiquitylation; Pif1-Rrm3
**Subject Category** DNA Replication, Recombination & Repair

## Introduction

The duplication of eukaryotic chromosomes initiates at many origins of DNA replication, each of which can only be activated once per cell cycle, to ensure that cells produce a single copy of each chromosome (Bell and Labib, 2016; Costa and Diffley, 2022). The initiation process involves the stepwise assembly of the DNA helicase known as CMG (Cdc45-MCM-GINS), which forms the core of the replisome at replication forks (Fig. EV1A). Firstly, two hexameric rings of the six Mcm2–7 ATPases are loaded around double-strand DNA (dsDNA) at origins during G1 phase, to produce inactive Mcm2–7 double hexamers (Evrin et al, 2009; Gambus et al, 2011; Remus et al, 2009). Once cells enter S-phase,

Mcm2–7 double hexamers at a subset of origins are converted by a set of initiation factors into pairs of CMG helicase complexes that initially encircle dsDNA (Douglas et al, 2018; Gambus et al, 2006; Lewis et al, 2022; Moyer et al, 2006; Wasserman et al, 2019; Yeeles et al, 2015). Subsequently, each pair of CMG complexes is activated in a step that requires the Mcm10 protein and involves the extrusion of one of the two parental DNA strands from the Mcm2–7 ring of each helicase (Douglas et al, 2018; Lewis et al, 2022). This generates two highly stable CMG complexes at the heart of a pair of bidirectional replisomes, with the Mcm2–7 ring of each CMG complex tracking in a 3′ to 5′ direction along single-strand DNA (ssDNA) that represents the template strand for leading-strand synthesis (Fu et al, 2011). CMG is essential to unwind the parental DNA duplex at replication forks and remains continuously associated with the DNA template throughout elongation (Kanemaki et al, 2003; Labib et al, 2000; Pacek and Walter, 2004; Tercero et al, 2000).

The Rrm3 DNA helicase and its paralogue Pif1 also act at DNA replication forks (Bochman et al, 2010; Malone et al, 2022; Muellner and Schmidt, 2020), helping forks to pass tightly bound protein-DNA complexes including Mcm2–7 double hexamers (Hill et al, 2020) and stable complexes at tRNA promoters or telomeres (Ivessa et al, 2003; Ivessa et al, 2002), as well as stable DNA structures such as G-quadruplexes (Kumar et al, 2021; Paeschke et al, 2011; Ribeyre et al, 2009; Varon et al, 2024; Williams et al, 2023). The activity of Rrm3 and Pif1 is stimulated by fork DNA, and they track in a 5′ to 3′ direction (Ivessa et al, 2002; Lahaye et al, 1993), enabling them to associate with the opposite strand at replication forks to CMG.

When two replication forks from neighbouring origins converge, Rrm3 and Pif1 help to unwind the final stretch of parental DNA between the two replisomes (Claussin et al, 2022; Deegan et al, 2019). The converging CMG helicases can then bypass each other (Low et al, 2020), since they encircle opposite strands of the parental DNA duplex. This breaks the association of each helicase with the DNA template for lagging strand synthesis, which was excluded from the Mcm2–7 ring at replication forks. By continuing to track along the leading-strand template, both helicases encounter the flapless end of the lagging strand of the opposing fork and can transition to encircling dsDNA (Fig. EV1A).

Unwinding the final stretch of parental DNA during termination provides a trigger for ubiquitylation and disassembly of the CMG helicase (Deegan et al, 2020; Dewar et al, 2015; Maric et al, 2014; Moreno et al, 2014). Before termination, the excluded DNA

[1]MRC Protein Phosphorylation and Ubiquitylation Unit, School of Life Sciences, University of Dundee, Dundee DD1 5EH, UK. [2]MRC Human Genetics Unit, Institute of Genetics and Cancer, University of Edinburgh, Edinburgh EH4 2XU, UK. ✉E-mail: t.deegan@ed.ac.uk; kpmlabib@dundee.ac.uk

strand at replication forks blocks the association of the ubiquitin ligase SCF$^{Dia2}$ with a binding site on Mcm2–7 that is essential for the ubiquitylation of CMG (Deegan et al, 2020; Jenkyn-Bedford et al, 2021; Low et al, 2020). Thereby, CMG ubiquitylation is restricted to termination and represents the final regulated step during the completion of eukaryotic chromosome duplication. SCF$^{Dia2}$ promotes the conjugation of a long K48-linked ubiquitin chain on the Mcm7 subunit of CMG (Deegan et al, 2020; Maric et al, 2014), leading to recruitment of the Cdc48 ATPase via its ubiquitin receptors Ufd1 and Npl4 (Maric et al, 2017). Cdc48 then unfolds ubiquitylated Mcm7 (Deegan et al, 2020), triggering disintegration of the CMG helicase and thus inducing replisome disassembly.

Failure to disassemble the highly stable CMG helicase during DNA replication termination would leave the Mcm2–7 ring topologically trapped around the DNA duplex, representing a roadblock to a variety of chromatin transactions, including the progression and convergence of DNA replication forks in the subsequent cell cycle. Indeed, budding yeast cells lacking the Dia2 substrate receptor of SCF$^{Dia2}$ are highly sick, have a high rate of genome instability, and are inviable at lower growth temperatures (Blake et al, 2006; Koepp et al, 2006; Morohashi et al, 2009; Pan et al, 2006). Structure-guided mutations in the leucine-rich repeats of Dia2, which bind to the Mcm2–7 component of CMG (Jenkyn-Bedford et al, 2021), phenocopy the CMG disassembly and cold-sensitivity phenotypes of dia2Δ cells (Jenkyn-Bedford et al, 2021). However, Dia2 is also likely to use its leucine-rich repeats to bind other substrates, by analogy with other F-box proteins (Harper and Schulman, 2021). Until now, therefore, it was unclear whether the phenotypes of dia2Δ cells reflect the consequences of leaving 'old' CMG complexes on chromosomes after DNA replication termination, or else result from failure to ubiquitylate other potential substrates of SCF$^{Dia2}$, such as the silencing factor Sir4 (Burgess et al, 2012), the transcription factor Tec1 (Bao et al, 2004), the licensing factor Cdc6 (Kim et al, 2012), the replisome components Ctf4 and Mrc1 (Deegan et al, 2020; Mimura et al, 2009), or the recombination factor Rad51 (Antoniuk-Majchrzak et al, 2023).

In cells lacking Dia2, the CMG helicase persists on chromosomes from DNA replication termination until G1 phase of the next cell cycle (Maric et al, 2014). Old CMG complexes can still associate with other replisome components and are fully functional for CMG disassembly, even during G1 phase, should Dia2 be re-expressed (Maric et al, 2014). Subsequently, the assembly of new CMG complexes during S-phase is comparable in dia2Δ to wild-type cells. However, the CMG helicase does not continuously accumulate from one cell cycle to the next in cultures of dia2Δ cells and the total amount of CMG during S-phase is similar between dia2Δ and control cells (Maric et al, 2014). This suggests that the 'old' CMG complexes that persist from the previous cell cycle into G1 phase in dia2Δ cells are disassembled during the subsequent S-phase, by a yet unknown pathway (Fig. EV1B).

In metazoa, CMG ubiquitylation and disassembly is induced by the ubiquitin ligase CUL2$^{LRR1}$ during DNA replication termination (Dewar et al, 2017; Fan et al, 2021; Le et al, 2021; Sonneville et al, 2017; Villa et al, 2021) and by the TRAIP ubiquitin ligase during mitosis (Deng et al, 2019; Sonneville et al, 2019). Like yeast SCF$^{Dia2}$, CUL2$^{LRR1}$ also ubiquitylates the MCM7 subunit of CMG, in a manner that is repressed at replication forks by the parental DNA

strand that is excluded from the Mcm2–7 ring (Jenkyn-Bedford et al, 2021), thereby restricting CMG-MCM7 ubiquitylation to termination. However, metazoan CUL2$^{LRR1}$ evolved independently to fungal SCF$^{Dia2}$, and neither enzyme appears to have orthologues in plant genomes. Similarly, orthologues of TRAIP are only found in metazoa. This indicates that CMG ubiquitylation arose more than once during eukaryotic evolution, raising the important question of how CMG disassembly was regulated in ancestral eukaryotes before the evolution of CMG ubiquitylation. Until now, it was unclear whether modern eukaryotes have additional pathways for CMG helicase disassembly that are independent of ubiquitylation and involve factors that evolved before the split between opisthokonts and modern-day plants.

To begin to address all these questions, we developed a mutated allele of Mcm7 that is resistant to ubiquitylation by SCF$^{Dia2}$. Analysis of the new mcm7 allele indicates that defects in CMG ubiquitylation underlie the major phenotypes of cells lacking Dia2 and lead to genome instability in the subsequent cell cycle. Our data further indicate that the Pif1 family of DNA helicases act during S-phase to mitigate the impact of old CMG helicase complexes from the previous cell cycle, via a second pathway of CMG helicase disassembly that is essential for cell viability in the absence of Mcm7 ubiquitylation. This suggests that Pif1 helicases might have driven an ancient pathway of CMG disassembly in ancestral eukaryotes, predating the evolution of CMG helicase ubiquitylation.

# Results

## Mutation of amino-terminal lysines in Mcm7 impairs ubiquitylation and disassembly of the CMG helicase

To explore the consequences of directly impairing CMG ubiquitylation in budding yeast cells, we attempted to develop a form of the CMG helicase that was poorly ubiquitylated by SCF$^{Dia2}$ and Cdc34. This was challenging, since ubiquitin conjugating enzymes (E2 enzymes) do not target specific sites within a consensus sequence, but instead are positioned by ubiquitin ligases (E3 enzymes) in close proximity to their substrates, and thus are able to access a range of potential ubiquitylation sites.

Previously, we showed that the first lysine in budding yeast Mcm7, namely Mcm7-K29 (Fig. EV2A,B), is the sole site of in vitro CMG ubiquitylation by SCF$^{Dia2}$, when CMG is released from replication fork DNA into budding yeast cell extracts (Maric et al, 2017). However, Mcm7-K29 is dispensable in vivo for CMG ubiquitylation in budding yeast cells (Maric et al, 2017). This suggested that CMG ubiquitylation is more efficient in vivo than in yeast cell extracts. Consistent with past findings (Deegan et al, 2020), we reconstituted the ubiquitylation of recombinant CMG (Fig. EV3A) with purified proteins and found that mutation of Mcm7-K29 blocked CMG ubiquitylation, under conditions where SCF$^{Dia2}$ (E3) and Cdc34 (E2) supported the conjugation of up to about 12 ubiquitins to CMG-Mcm7 (Fig. EV3B, lanes 5–6). In contrast, CMG-Mcm7-K29A was robustly ubiquitylated under more efficient conditions that allowed SCF$^{Dia2}$ and Cdc34 to conjugate much longer ubiquitin chains to CMG-Mcm7, though ubiquitylation of CMG-Mcm7-K29A was still less efficient than wild-type CMG (Fig. EV3B, lanes 7–8). These data showed that

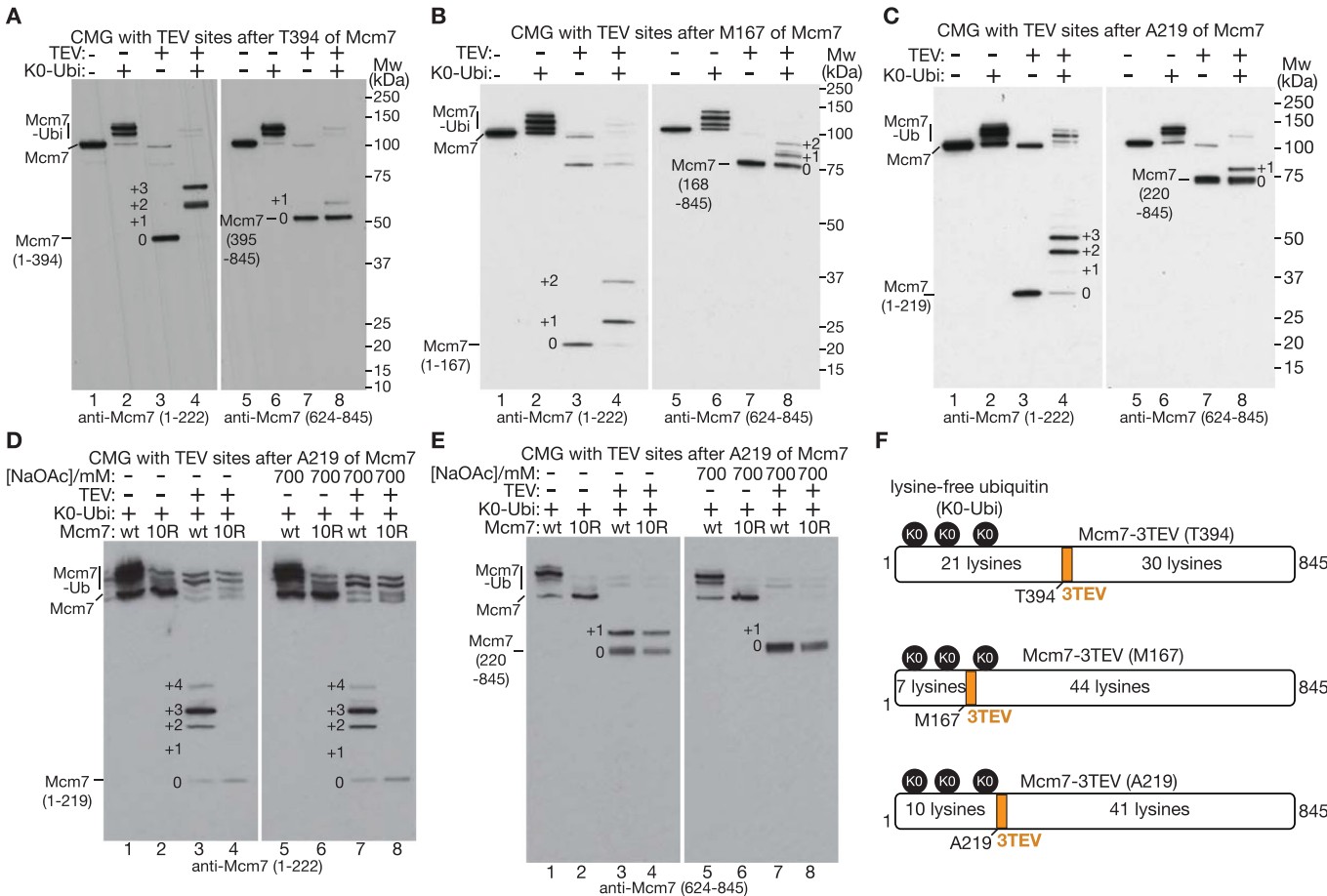

**Figure 1. CMG ubiquitylation by SCF^Dia2 and Cdc34 has a strong preference for the amino-terminal region of Mcm7.**

(A) Budding yeast CMG with three TEV sites inserted after Mcm7-T394 was ubiquitylated in reactions containing lysine-free ubiquitin (K0 Ubi) as indicated. The reaction products were then cleaved with TEV protease as shown and analysed by immunoblotting with antibodies specific to the amino terminus of Mcm7 (1–222) or the carboxyl terminus (624–845). Where necessary, the identity of the various ubiquitylated forms was determined via longer exposures of the same immunoblot (e.g. for the +1 form in lane 4). (B, C) Equivalent analysis for CMG with three TEV sites inserted after Mcm7-M167 (B) or Mcm7-A219 (C). (D) Reconstituted CMG ubiquitylation reactions were stopped as indicated by increasing the salt concentration (700 mM NaOAc), before cleavage of TEV sites inserted after A219 of Mcm7 (see Methods for details). Reactions were analysed by immunoblotting with antibodies to Mcm7 1–222. (E) The reactions in (D) were also analysed by immunoblotting with antibodies to Mcm7 624–845. (F) Summary of lysine distribution and observed ubiquitylation in Mcm7, either side of TEV cleavage sites inserted after M167, A219 or T394 (3TEV = three consecutive TEV cleavage sites). Source data are available online for this figure.

SCF^Dia2 and Cdc34 can ubiquitylate additional sites on CMG-Mcm7 that still remained to be mapped.

Given the failure of mass spectrometry analysis of in vivo ubiquitylated CMG-Mcm7 to reveal additional sites of ubiquitylation (Maric et al, 2017), we took an alternative approach that was based on the insertion of cleavage sites for the Tobacco Etch Virus (TEV) protease into three poorly conserved disordered loops in yeast Mcm7 (Fig. EV2A,B; Movie EV1). Ubiquitylation of CMG containing such variants, followed by TEV cleavage, should identify regions of Mcm7 that contain ubiquitylation sites, which could then be characterised further by mutational analysis. Recombinant CMG complexes containing each of the three TEV-site insertions in Mcm7 were purified (Fig. EV3C) and shown to be ubiquitylated by SCF^Dia2 and Cdc34 with comparable efficiency to wild-type CMG in reconstituted in vitro reactions (Fig. EV3D,E). Subsequently, reactions were repeated under the highly efficient conditions shown in Fig. EV3B but using lysine-free ubiquitin (K0 Ubi), to

block the formation of ubiquitin chains and reveal how many sites were being modified in each molecule of Mcm7. Ubiquitylated CMG complexes were cleaved with TEV protease and the resulting fragments of Mcm7 were then analysed by immunoblotting, with antibodies specific to the amino-terminal or carboxy-terminal regions of Mcm7 (Fig. 1A–C).

Using CMG with TEV sites inserted after T394 of Mcm7, ubiquitin was conjugated to up to three sites per molecule within the first 394 amino acids of Mcm7 (Fig. 1A, lanes 3–4; note that the assay does not show whether the same three sites are modified in each Mcm7 molecule). In contrast, Mcm7 ubiquitylation downstream of T394 was very inefficient (Fig. 1A, lanes 7–8). These data indicate that SCF^Dia2 and Cdc34 have a strong preference for ubiquitylating the amino-terminal half of Mcm7 (Fig. 1F, Mcm7-3TEV (T394)), consistent with previous work showing that Mcm7-K29 is a favoured site (Maric et al, 2017). Subsequently, we analysed cleavage of ubiquitylated CMG complexes with TEV sites inserted

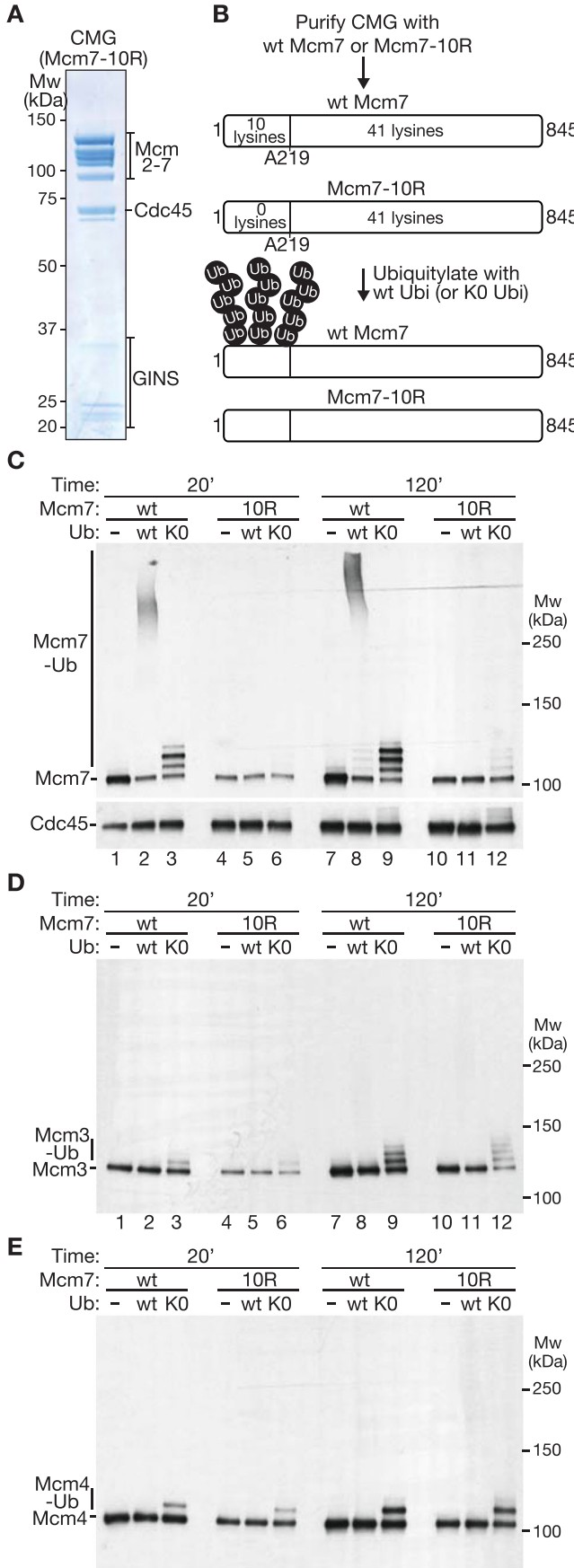

Figure 2. Mcm7-10R impairs in vitro ubiquitylation of the CMG helicase by SCF^Dia2 and Cdc34.

(A) Purified recombinant budding yeast CMG helicase with the Mcm7-10R allele. (B) Scheme illustrating the ubiquitylation of CMG with wild-type Mcm7 (Fig. EV3A) or Mcm7-10R (A). (C) After ubiquitylation of CMG and CMG-Mcm7-10R under the indicated conditions, modification of Mcm7 was monitored by immunoblotting. Cdc45 was included as a loading control. (D, E) Ubiquitylation of Mcm3 and Mcm4 was monitored in the same experiment as above. Source data are available online for this figure.

after M167 of Mcm7 (Fig. 1B). In this case, up to two ubiquitins were conjugated with high efficiency to the first 167 amino acids, whereas ubiquitylation downstream of M167 was less efficient but still involved up to two ubiquitins per Mcm7 molecule. Combined with the above data, these findings indicated the presence of at least two ubiquitylation sites in Mcm7 1–167, plus up to two sites in the remainder of the protein (Fig. 1F, Mcm7-3TEV (M167)). Finally, we examined the cleavage products of CMG with TEV sites inserted after A219 of Mcm7 (Fig. 1C) and saw that 2–3 ubiquitins were conjugated to the amino-terminal fragment after TEV cleavage (Fig. 1C, lanes 3–4), together with inefficient conjugation of a single ubiquitin downstream of the cleavage sites. In summary, these data indicated that the first 219 amino acids of Mcm7 contain the major sites at which the CMG helicase is ubiquitylated by SCF^Dia2 and Cdc34 (Fig. 1F, Mcm7-3TEV (A219)).

Finally, we tested whether any of the observed ubiquitylation sites in Mcm7 were induced by TEV cleavage, which might increase the access of SCF^Dia2 and Cdc34 to residues around the cleavage site. After ubiquitylating CMG-Mcm7(A219-TEV) as above, the sample was split in two, and the salt concentration increased in one half to stop the ubiquitylation reaction, before cleavage of Mcm7 with TEV (Fig. 1D). Ubiquitylation of the first 219 amino acids of Mcm7 was unaffected by the presence or absence of high salt during TEV cleavage (Fig. 1D, compare lanes 3 and 7), with 2–4 ubiquitins being conjugated to each Mcm7 molecule in both cases. In contrast, a single ubiquitin was conjugated to Mcm7 downstream of A219 in the control sample (Fig. 1E, lane 3), but this was blocked by increasing the salt concentration (Fig. 1E, lane 7). These data confirmed the presence of multiple ubiquitylation sites within the first 219 amino acids of Mcm7 and further indicated that TEV cleavage leads to inefficient and artefactual ubiquitylation downstream of the cleavage site. Ubiquitylation of the amino-terminal region of CMG-Mcm7 was restricted to lysine residues, since mutation of all 10 lysines in the first 219 amino acids to arginine (Fig. EV3C, Mcm7-10R(A219-TEV)) was sufficient to block ubiquitylation of the Mcm7 amino-terminal fragment after TEV cleavage (Fig. 1D, lane 4). These lysine residues are located on the surface of Mcm7, in a region likely to be accessible to Cdc34 when bound to the RING domain of SCF^Dia2 (Fig. EV2C, D; Movie EV2). Mutation of ubiquitylation sites in Mcm7 did not lead to enhanced ubiquitylation of Mcm2–6 (Fig. EV3F).

Based on these findings, a version of CMG containing Mcm7-10R but lacking TEV cleavage sites was purified and compared in ubiquitylation reactions to wild-type CMG (Fig. 2A,B). Under conditions where wild-type CMG was ubiquitylated robustly (Fig. 2C, lanes 2 and 8), on up to four sites per Mcm7 molecule (Fig. 2C, lane 9), the ubiquitylation of CMG-Mcm7-10R was almost undetectable in reactions containing wild-type ubiquitin (Fig. 2C,

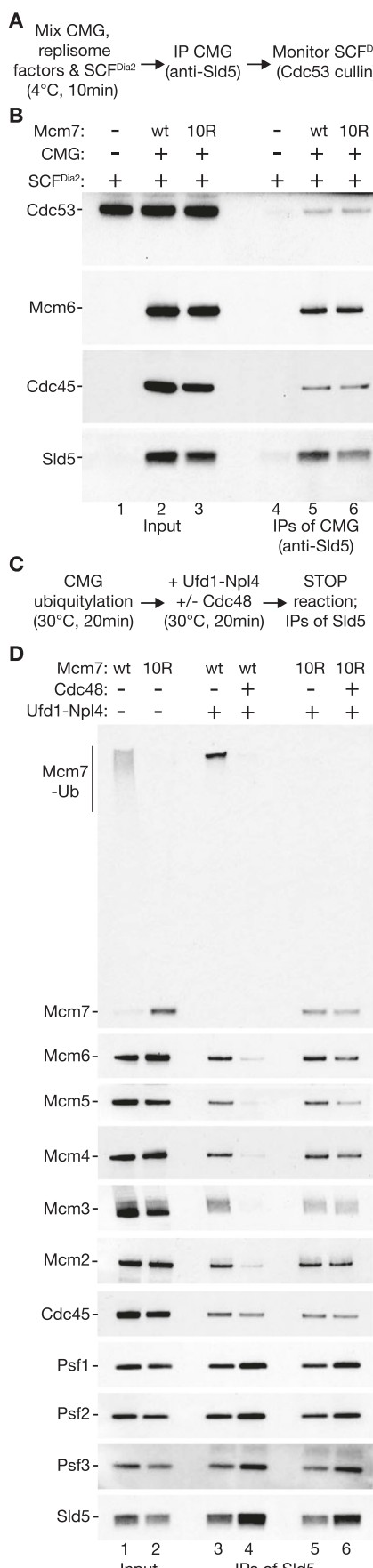

**A** Mix CMG, replisome factors & SCF^Dia2 (4°C, 10min) → IP CMG (anti-Sld5) → Monitor SCF^Dia2 (Cdc53 cullin)

**B**
Mcm7: −  wt  10R    −  wt  10R
CMG: −  +  +    −  +  +
SCF^Dia2: +  +  +    +  +  +

Cdc53 —
Mcm6 —
Cdc45 —
Sld5 —

1  2  3   4  5  6
Input    IPs of CMG (anti-Sld5)

**C** CMG ubiquitylation (30°C, 20min) → + Ufd1-Npl4 +/− Cdc48 (30°C, 20min) → STOP reaction; IPs of Sld5

**D**
Mcm7: wt  10R   wt  wt   10R  10R
Cdc48: −  −    −  +    −  +
Ufd1-Npl4: −  −    +  +    +  +

Mcm7-Ub —
Mcm7 —
Mcm6 —
Mcm5 —
Mcm4 —
Mcm3 —
Mcm2 —
Cdc45 —
Psf1 —
Psf2 —
Psf3 —
Sld5 —

1  2   3  4   5  6
Input    IPs of Sld5

**Figure 3. Mcm7-10R associates with SCF^Dia2 in vitro but blocks CMG disassembly.**

(A) Reaction scheme to monitor association of budding yeast CMG (wt Mcm7 or Mcm7-10R) with SCF^Dia2 in the presence of the replisome factors Ctf4, Mrc1, DNA polymerase epsilon. (B) For the reactions described in (A), the indicated factors were monitored by immunoblotting. (C) Scheme for assaying the disassembly of ubiquitylated CMG by Cdc48-Ufd1-Npl4. (D) The indicated factors were monitored by immunoblotting, for the reactions described in (C). Source data are available online for this figure.

lanes 5 and 11), though reactions containing lysine-free ubiquitin revealed inefficient ubiquitylation of Mcm7-10R (Fig. 2C, lane 12; note that lysine-free ubiquitin provides a more sensitive readout as the products do not smear up the gel lane). Similarly, lysine-free ubiquitin revealed inefficient ubiquitylation of the Mcm3 and Mcm4 subunits of CMG as reported previously (Deegan et al, 2020; Maric et al, 2017; Mukherjee and Labib, 2019), regardless of the presence of the 10R mutations in Mcm7 (Fig. 2D,E). In summary, these data indicate that the Mcm7-10R allele greatly reduces CMG ubiquitylation without abolishing it completely.

To confirm that the ubiquitylation defect of Mcm7-10R is due to loss of ubiquitylation sites, without affecting the association of CMG-Mcm7-10R with SCF^Dia2, we incubated wild-type CMG or CMG-Mcm7-10R with SCF^Dia2, in the presence of Ctf4, Mrc1 and DNA polymerase epsilon that help to link the ubiquitin ligase to the helicase (Deegan et al, 2020; Maculins et al, 2015; Morohashi et al, 2009). We then isolated CMG by immunoprecipitation of the Sld5 subunit of the GINS component (Fig. 3A). This showed that SCF^Dia2 interacted equally well with wild-type CMG and CMG-Mcm7-10R (Fig. 3B).

To demonstrate that the ubiquitylation defect of Mcm7-10R impairs CMG helicase disassembly, we performed ubiquitylation reactions with wild-type CMG and CMG-Mcm7-10R as above. Purified recombinant versions of Cdc48 and its adaptor complex Ufd1-Npl4 were then added to the reaction mixtures for 20 min. Subsequently, the reactions were stopped by increasing the salt concentration before incubation with beads coated with antibodies to the Sld5 subunit of GINS (Fig. 3C). In reactions containing wild-type CMG, addition of Cdc48-Ufd1-Npl4 disrupted the interaction between GINS and the Mcm2–7 complex (Fig. 3D, compare lanes 3–4), indicating efficient helicase disassembly as described previously (Deegan et al, 2020). In contrast, the disassembly of CMG-Mcm7-10R was greatly impaired (Fig. 3D, compare lanes 5–6).

## Phenotypic analysis of *mcm7-10R* indicates that CMG is a major target of SCF^Dia2 in the budding yeast cell cycle

To explore the phenotypic consequences of impairing CMG ubiquitylation and disassembly, the *mcm7-10R* mutations were introduced into the *mcm7* locus in budding yeast cells. Mcm7-10R cells were viable and replicated with similar kinetics to control cells (Fig. 4A,D), indicating that the mutation of surface lysines did not impair the assembly or action of the CMG helicase. To examine the impact of *mcm7-10R* on CMG ubiquitylation in vivo, we used cells in which Cdc48 was fused to the auxin-inducible degron or 'AID' (Maric et al, 2014; Nishimura et al, 2009), allowing us to degrade Cdc48-AID in early S-phase and thereby monitor the accumulation of ubiquitylated CMG during DNA replication termination.

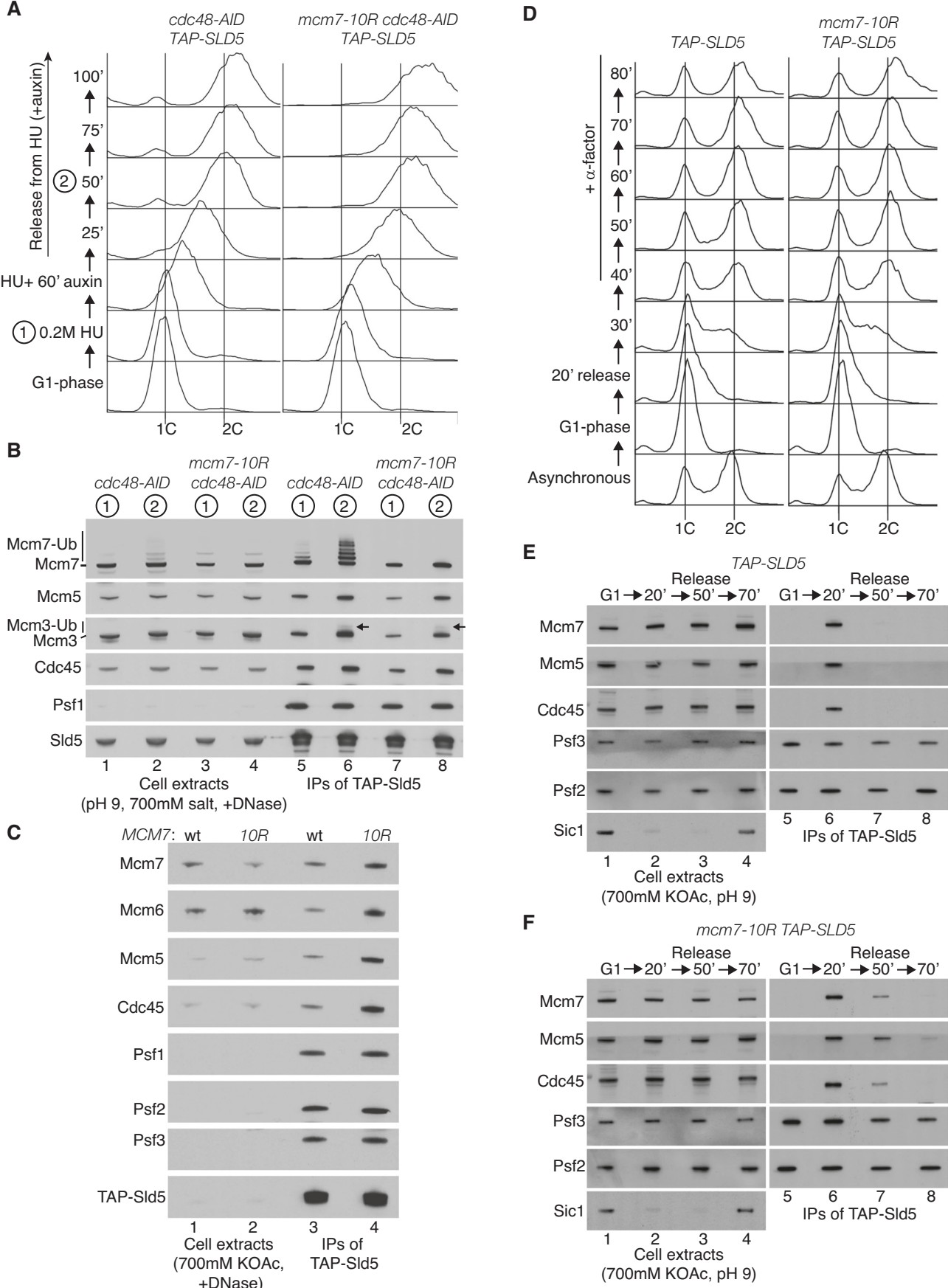

◄

**Figure 4.** *mcm7-10R* impairs CMG ubiquitylation and delays CMG disassembly in budding yeast cells.

(A) Budding yeast *cdc48-AID TAP-SLD5* (YPNK337) and *mcm7-10R cdc48-AID TAP-SLD5* (YCPR86) strains were arrested in G1 phase with mating pheromone at 30 °C and then released into S-phase for 60 minutes in the presence of 0.2 M hydroxyurea (HU). Subsequently, auxin (0.5 mM IAA) was added for 60 min in the continued presence of HU, and cells were then washed into fresh medium containing auxin lacking hydroxyurea, to allow progression through S-phase. DNA content was monitored throughout the experiment by flow cytometry. (B) Cell extracts were prepared at the indicated times for the experiment in (A) and used to isolate CMG via the TAP-tagged Sld5 subunit of GINS, under conditions that blocked in vitro CMG ubiquitylation. The indicated factors were monitored by immunoblotting. Arrows indicate the small population of ubiquitylated Mcm3 after inactivation of Cdc48-AID. (C) TAP-Sld5 was isolated from extracts of asynchronous cultures of control cells (YSS47) and *mcm7-10R* cells (YCPR79). (D) *TAP-SLD5* (YSS47) and *mcm7-10R TAP-SLD5* (YCPR79) strains were arrested in G1 phase with mating pheromone at 30 °C and then released into S-phase in fresh medium. Mating pheromone was added again after 40 min to prevent cells from entering S-phase in the subsequent cell cycle. (E, F) TAP-Sld5 was isolated from cell extracts at the indicated timepoints. Sic1 was also monitored in cell extracts as a marker for arrest in G1 phase. Source data are available online for this figure.

*cdc48-AID* and *cdc48-AID mcm7-10R* cells were released from G1 phase into early S-phase in the presence of 0.2 M hydroxyurea (Fig. 4A), to inhibit ribonucleotide reductase and slow the progression of replication forks from early origins, leading to repression of late origin firing via the S-phase checkpoint pathway. Subsequently, auxin was added for one hour to inactivate Cdc48-AID, and cells were then released into fresh medium lacking hydroxyurea, allowing them to complete chromosome replication. Samples were taken during the early and late S-phase (Fig. 4A, samples 1 and 2) and used to isolate the CMG helicase from cell extracts, by immunoprecipitation of a TAP-tagged version of the Sld5 subunit of GINS. When *cdc48-AID* control cells completed S-phase, CMG accumulated with ubiquitylated Mcm7 and residual ubiquitylation of Mcm3 (Fig. 4B, lane 6; note that ubiquitin chains are short under these conditions, likely due to partial depletion of free ubiquitin upon inactivation of Cdc48-AID). In contrast, ubiquitylation of CMG-Mcm7 was impaired by the *mcm7-10R* mutations, without affecting inefficient Mcm3 ubiquitylation (Fig. 4B, lane 8). These data demonstrated that *mcm7-10R* interferes with CMG ubiquitylation in budding yeast cells but does not block it entirely.

To monitor the impact of *mcm7-10R* on CMG disassembly, we first examined the level of the helicase in extracts of asynchronous cell cultures, by immunoprecipitation of TAP-tagged Sld5. A similar amount of GINS was isolated from control cells and *mcm7-10R*, but the association of GINS with Cdc45 and Mcm2–7 was enhanced in *mcm7-10R*, indicating accumulation of the CMG helicase (Fig. 4C). Subsequently, the assembly and disassembly of CMG were monitored in synchronised cultures of control and *mcm7-10R* cells, following arrest in G1 phase and release into S-phase (Fig. 4D). In control cells, CMG was detected 20 min after release from G1 arrest, corresponding to cells in early S-phase (Fig. 4E, lane 6). Thirty minutes later, CMG was scarcely detectable (Fig. 4E, lane 7), reflecting the fact that S-phase takes about 20 min in budding yeast. CMG assembly in *mcm7-10R* cells was comparable to the control (Fig. 4F, lanes 5–6). However, the disappearance of CMG was markedly slower in *mcm7-10R* cells (Fig. 4F, lane 7), indicating that CMG ubiquitylation was impaired but not completely blocked, in contrast to *dia2Δ* cells in which ubiquitylation and disassembly during a single cell cycle is blocked entirely (Maric et al, 2014). This suggests that CMG disassembly under such conditions is driven by residual ubiquitylation of Mcm7, Mcm3 or Mcm4 (c.f. Fig. 2C–E).

The impact of *mcm7-10R* on CMG ubiquitylation and disassembly is reminiscent of the effect of deleting the amino-terminal TPR domain of DIA2 (TPR—tetratricopeptide repeat). Previous work showed that the Dia2-TPR domain increases the

efficiency of CMG ubiquitylation by binding to Ctf4 and Mrc1 (Deegan et al, 2020; Maculins et al, 2015), which associate with the CMG helicase in the replisome (Gambus et al, 2006). In *dia2-ΔTPR* cells, CMG ubiquitylation is impaired to a lesser degree than in *mcm7-10R*, but the defect is still sufficient to delay CMG helicase disassembly (Maculins et al, 2015).

Since CMG disassembly is delayed but not abolished in *mcm7-10R* or *dia2-ΔTPR*, most helicase complexes were disassembled when cells were arrested in G1 phase for an extended period (Fig. 5A,B, lanes 6–7). In contrast, CMG persisted from DNA replication termination to the G1 phase of the following cell cycle in *dia2Δ* cells (Fig. 5A,B, lane 5), as shown previously (Maric et al, 2014). Notably, the CMG disassembly defects of *mcm7-10R* and *dia2-ΔTPR* are synergistic, and most CMG complexes persisted into G1 phase of the following cell cycle in the *mcm7-10R dia2-ΔTPR* double mutant (Fig. 5A,B, lane 8), as seen with *dia2Δ*. Moreover, *mcm7-10R dia2-ΔTPR* reproduced other previously reported phenotypes of *dia2Δ* cells (Blake et al, 2006; Koepp et al, 2006; Morohashi et al, 2009; Pan et al, 2006), such as cold-sensitivity (Fig. 5C), sensitivity to the alkylating agent methyl methanesulfonate (MMS, Fig. 5D) and synthetic lethality with *mec1Δ sml1Δ* or *slx8Δ* (Fig. EV4). In addition, the *mcm7-10R* single mutant was mildly sensitive to cold (Fig. 5C) and MMS (Fig. 5D). These findings suggested that failure to disassemble the CMG helicase is a major determinant of *dia2Δ* phenotypes. Consistent with this view, mass spectrometry analysis of Protein A-tagged Dia2, isolated from extracts of S-phase yeast cells, indicated that the replisome is the major partner of SCF$^{Dia2}$, largely dependent upon the TPR domain of Dia2 (Appendix Tables S1 and S2; Datasets EV1 and EV2). In contrast, other reported substrates of SCF$^{Dia2}$ such as Tec1, Rad51, Sir4 and Cdc6, which have not been found to associate with CMG, were either not detected or else were very weakly enriched, in mass spectrometry analysis of immunoprecipitates of Protein A-tagged Dia2 (Appendix Tables S1 and S2; Datasets EV1 and EV2).

Cells lacking Dia2 accumulate sub-nuclear foci of the recombination factor Rad52 (Fig. 5E,F), indicating endogenous DNA damage (Blake et al, 2006; Morohashi et al, 2009). Whereas the proportion of cells with Rad52 foci was not increased significantly in *dia2-ΔTPR* cells, the proportion of *mcm7-10R* cells with Rad52 foci was intermediate between control cells and *dia2Δ* (Fig. 5F), reflecting the stronger CMG ubiquitylation defect in *mcm7-10R* compared to *dia2-ΔTPR*. The proportion of cells with Rad52 foci increased further in the *mcm7-10R dia2-ΔTPR* double mutant, to a level that was indistinguishable from *dia2Δ*. Together with the above analysis, and the previous characterisation of structure-guided mutations in the leucine-rich repeats of Dia2 (Jenkyn-Bedford et al, 2021), these data indicate that CMG is a major target

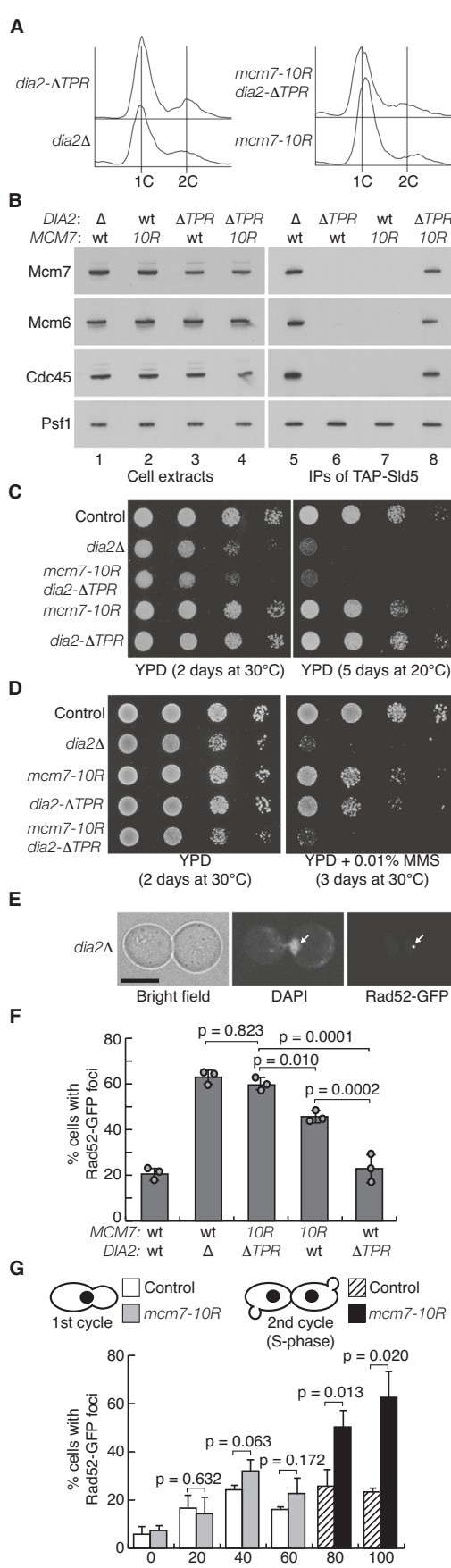

 **Figure 5.** Defects in CMG helicase disassembly phenocopy loss of Dia2 and induce genome instability in the next cell cycle.

(A) Budding yeast cells of the indicated genotypes (YHM130, YTM265, YCPR86, YCPR111) were arrested in G1 phase at 30 °C by addition of mating pheromone. (B) GINS and the CMG helicase were isolated from cell extracts by immunoprecipitation of TAP-tagged Sld5. The indicated factors were monitored by immunoblotting. (C, D) Serial dilutions of the indicated genotypes (YHM28, YHM306, YCPR68, YCPR103 were spotted on media under the indicated conditions. YPD 'Yeast extract Peptone Dextrose' medium, MMS methyl methanesulfonate. (E) Example of a *dia2Δ* cell with a sub-nuclear focus of Rad52-GFP (marked by white arrows). The scale bar corresponds to 5 µm. (F) The percentage of cells with sub-nuclear foci of Rad52-GFP was monitored for the indicated genotypes (YBH295, YTM115, YCPR447, YCPR445, YTM74). The data from three independent experiments are shown as circles, with the histograms showing mean values and lines indicating standard deviation. Statistical analysis was performed using a one-way ANOVA test followed by Tukey's test, to generate the indicated *P* values. (G) Control (YBH295) and *mcm7-10R* (YCPR445) were arrested in the G1 phase by addition of mating pheromone before release into fresh medium lacking mating pheromone. The proportion of cells with sub-nuclear foci of Rad52-GFP was quantified at the indicated timepoints, for cells in the first cell cycle or second cell cycle. The latter were identified as cells that had completed nuclear division and formed a small bud (the cartoon illustrates that cells budded before the completion of cell separation, indicating that G1 phase was very short, likely due to continued cell growth in the original G1 arrest). The histograms indicate the mean values from three independent experiments, with standard deviation indicated by lines. Statistical analysis was performed using a two-way ANOVA test followed by Tukey's test, to generate the indicated *P* values. Source data are available online for this figure.

of SCF^Dia2 in the budding yeast cell cycle, with the persistence of 'old' CMG helicase complexes being a major driver for the phenotypes of cells lacking Dia2.

## Old CMG complexes increase the frequency of Rad52 recombination foci in the next cell cycle

As noted above, the persistence of old CMG complexes in *mcm7-10R* cells can be largely suppressed by arresting cells for an extended period in the G1 phase of the subsequent cell cycle, by the addition of mating pheromone to the cell culture (Fig. 5B, lane 7). This enabled us to compare how defective CMG disassembly drives genome integrity in the first and second cell cycles, by releasing control cells and *mcm7-10R* cells from an extended G1 arrest and then monitoring the kinetics of Rad52 sub-nuclear foci, as cells progressed through the first cell cycle and then entered the second. Cells continued to grow and increase their size during the initial arrest in G1 phase, leading subsequently to rapid entry into S-phase of the first cell cycle in fresh medium lacking mating pheromone, and a very short G1 phase in the second cell cycle.

Upon release into S-phase of the first cell cycle, the proportion of cells with sub-nuclear foci of Rad52-GFP was initially similar in control cells and *mcm7-10R* (Fig. 5G, 0–20 min), consistent with *mcm7-10R* cells only containing a residual level of old CMG complexes when they entered S-phase. However, when cells completed the first cell cycle and rapidly entered the following cycle, the proportion of cells with sub-nuclear foci of Rad52-GFP was greatly enhanced in *mcm7-10R* cells (Fig. 5G, 80–100 min; *mcm7-10R* 2nd cycle). These findings indicated that old CMG complexes from one cell cycle represent a source of genome instability when cells enter S-phase of the subsequent cell cycle.

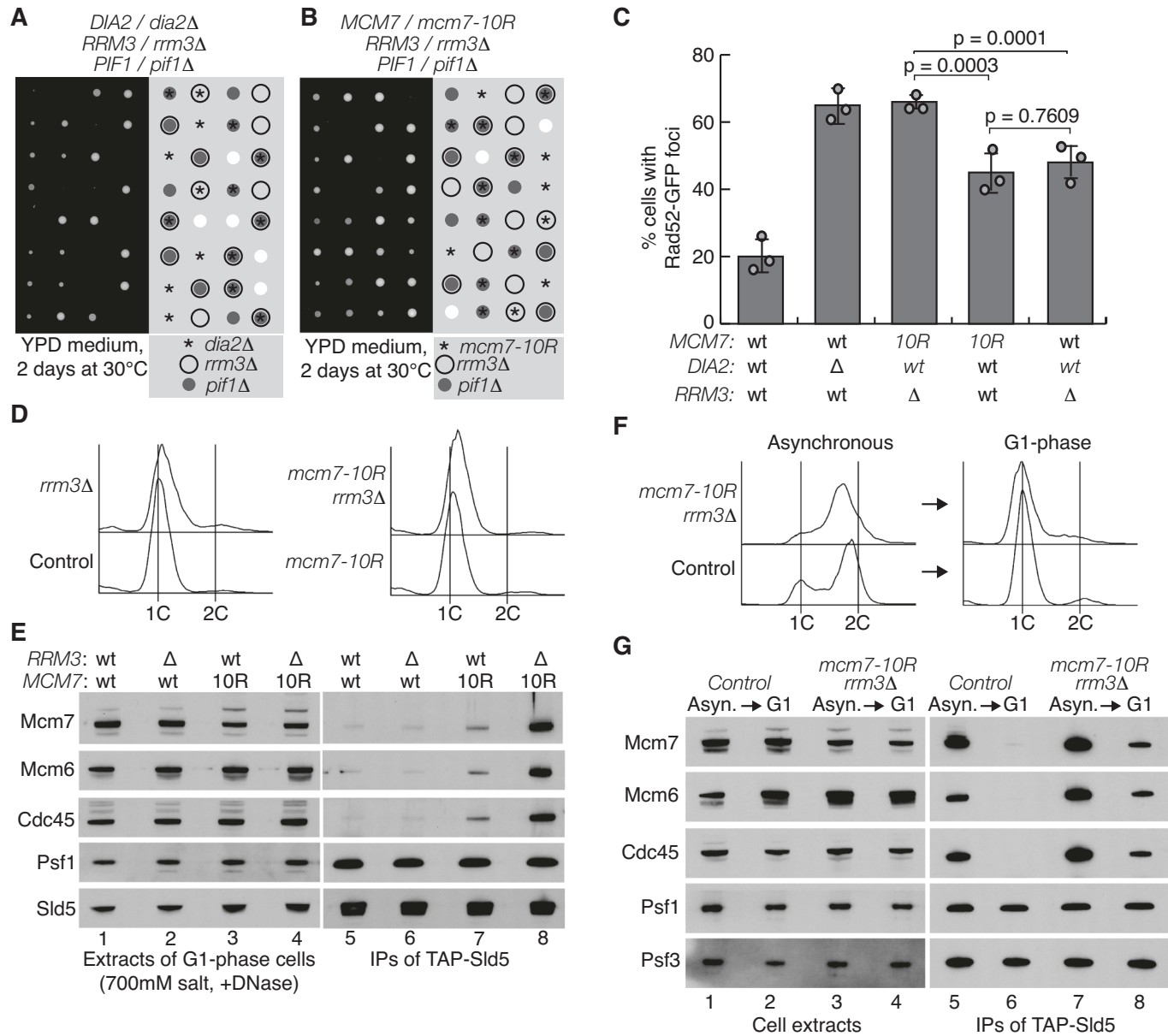

Figure 6. Pif1-family helicases are essential for viability and important for CMG helicase disassembly when Mcm7 ubiquitylation is defective.

(A, B) Tetrad analysis for diploid budding yeast cells of the indicated genotypes. YPD = medium based on Yeast Extract, Peptone, and Dextrose. (C) The percentage of cells with sub-nuclear foci of Rad52-GFP was monitored for the indicated genotypes (YBH295, YTM115, YCPR419, YCPR445, YTM120). The data from three independent experiments are shown as circles, with the histograms showing mean values and lines indicating standard deviation. Statistical analysis was performed using a one-way ANOVA test followed by Tukey's test, to generate the indicated P values. (D) The indicated budding yeast strains (YSS47, YTM371, YCPR79 and YCPR143) were grown at 30 °C and arrested in G1 phase by the addition of mating pheromone. DNA content was monitored by flow cytometry. (E) Cell extracts were prepared from the samples in (C) and used to isolate the CMG helicase by immunoprecipitation of TAP-Sld5. The indicated factors were monitored by immunoblotting. (F) Control cells (YSS47 and mcm7-10R rrm3Δ (YCPR143) were grown asynchronously at 30 °C and then arrested in G1 phase as above. (G) GINS and associated factors were isolated from cell extracts by immunoprecipitation of TAP-Sld5 and then monitored by immunoblotting. Source data are available online for this figure.

## Pif1-family helicases are essential for the viability of dia2Δ and mcm7-10R and mediate a second pathway for CMG helicase disassembly

As discussed above, old CMG complexes from the previous cell cycle in dia2Δ cells are likely removed during S-phase by a previously unknown pathway (Fig. EV1B). Consistent with past

work (Blake et al, 2006; Morohashi et al, 2009), both Rrm3 and Pif1 are important for the viability of dia2Δ cells (Fig. 6A, both dia2Δ rrm3Δ and dia2Δ pif1Δ are extremely sick), suggesting a role for these helicases in such a pathway. In addition, we found that dia2Δ cells lacking both Rrm3 and Pif1 are inviable (Fig. 6A).

Similarly, the mcm7-10R rrm3Δ pif1Δ triple mutant is also inviable (Fig. 6B), and the proportion of mcm7-10R rrm3Δ cells

with Rad52-GFP foci was comparable to *dia2Δ* (Fig. 6C). In contrast, other replication fork associated helicases such as Chl1 (Samora et al, 2016; Skibbens, 2004; Srinivasan et al, 2020), Srs2 (Arbel et al, 2020; Crickard and Greene, 2019; Lehmann et al, 2020) and the Sgs1 orthologue of human BLM helicase (Gupta and Schmidt, 2020; Simmons et al, 2021) were not required for the viability of *mcm7-10R* (Fig. EV5B,D). These findings indicate that Pif1-family helicases are important for cell viability in response to defects in CMG helicase disassembly.

To explore whether Rrm3 and Pif1 contribute to the disassembly of old CMG complexes, in cells in which the ubiquitylation of CMG-Mcm7 is defective, we initially tried to use degron technology to generate a conditional version of the lethal *mcm7-10R rrm3Δ pif1Δ* or *dia2Δ rrm3Δ pif1Δ* strains. However, this was not successful due to incomplete degradation of degron-tagged protein, and so we focussed on characterising the viable combination of *mcm7-10R* with *rrm3Δ*, after arresting cells in G1 phase (Fig. 6D). This revealed a striking persistence during G1 phase of the CMG helicase in the *mcm7-10R rrm3Δ* double mutant (Fig. 6E, lane 8), compared to control cells (Fig. 6E, lane 5), or to the *rrm3Δ* or *mcm7-10R* single mutants (Fig. 6E, lanes 6–7).

Only a proportion of CMG helicase complexes persisted into G1 phase in the *mcm7-10R rrm3Δ* mutant (Fig. 6F,G), either due to residual ubiquitylation of CMG-Mcm7-10R by SCF^Dia2, or else reflecting the ability of additional factors to contribute to CMG helicase disassembly. Pif1 is a likely candidate, especially since it is essential for the viability of *mcm7-10R rrm3Δ* cells (Fig. 6B). However, old CMG complexes did not persist during G1 phase in *mcm7-10R pif1Δ* cells (Fig. EV5E,F), suggesting that the role of Pif1 is likely masked to some degree by the dominant role of Rrm3 in this pathway, analogous to the major role of Rrm3 in unwinding the final stretch of parental DNA at converged replication forks (Claussin et al, 2022; Deegan et al, 2019). In addition, CMG did not persist into G1 phase when *mcm7-10R* was combined with *chl1Δ*, *sgs1Δ* or *srs2Δ* (Fig. EV5G–I). These findings highlight the dominant role of Rrm3 in the disassembly of old CMG helicase complexes that have not been removed via Mcm7 ubiquitylation and Cdc48.

## Rrm3 acts during S-phase to trigger the disassembly of old CMG complexes from the previous cell cycle

To determine when Rrm3 acts during the cell cycle, to disassemble old CMG helicase complexes that have not been ubiquitylated by SCF^Dia2, we generated *mcm7-10R rrm3Δ* cells that expressed the *RRM3* coding sequence at the *leu2* locus, under control of the regulatable *GAL1,10* promoter. Subsequently, *mcm7-10R rrm3Δ GAL-RRM3* cells were grown in the absence of Rrm3, in medium containing raffinose as the carbon source, before arresting in G1 phase by the addition of mating pheromone. The culture was then split in two and one half switched to medium containing galactose and mating pheromone, to induce expression of *GAL-RRM3* for 60 min whilst maintaining arrest in G1 phase. Subsequently, the cultures were washed into fresh medium lacking mating pheromone, to allow cells to enter S-phase. Forty minutes later, mating pheromone was added once again, to arrest cells in G1 phase of the next cell cycle (Fig. 7A). DNA content was monitored by flow cytometry throughout the experiment (Fig. 7B), and samples corresponding to the first and second G1-phases were used to prepare cell extracts and monitor the presence of old CMG helicase complexes, by immunoprecipitation of TAP-tagged Sld5 (Fig. 7C).

Induction of *GAL-RRM3* during G1 phase did not reduce the level of old CMG complexes in *mcm7-10R rrm3Δ GAL-RRM3* cells (Fig. 7C, compare lanes 5–6). In contrast, when Rrm3 was present upon release from G1 phase and throughout the ensuing cell cycle, most old CMG complexes were disassembled (Fig. 7C, compare lanes 7–8). This indicated that Rrm3 cannot drive the disassembly of old CMG complexes from G1 phase and instead acts at some later point in the cell cycle.

To test whether Rrm3 acts during S-phase to disassemble old CMG complexes from the previous cell cycle, the above experiment with *mcm7-10R rrm3Δ GAL-RRM3* cells was repeated, except that samples were taken at 10-min intervals upon release from G1 phase (Fig. 8A,B). In the absence of Rrm3 expression, the level of CMG increased 20 min after release from G1 phase and then persisted throughout S-phase, reflecting the addition of new CMG helicase complexes to the pool of old CMG from the previous cell cycle (Fig. 8C, compare lanes 5–8), under conditions where CMG ubiquitylation was defective (Fig. 9A, *GAL-RRM3* **OFF** throughout experiment). The level of CMG also increased when *mcm7-10R rrm3Δ GAL-RRM3* cells entered S-phase in the presence of Rrm3 (Fig. 8D, lanes 5–7). However, the amount of the CMG helicase peaked at 30 min and then declined 10 min later (Fig. 8D, compare lanes 7–8). These data indicate that Rrm3 acts during S-phase to remove old CMG helicase complexes from the previous cell cycle. At the same time, new CMG complexes are assembled and then subsequently persist after DNA replication termination, due to the ubiquitylation defect of *mcm7-10R* cells (Fig. 9A, *GAL-RRM3* switched **ON** during G1 phase). In summary, these findings show that budding yeast cells have two pathways for CMG helicase disassembly, one of which works during DNA replication termination and involves ubiquitylation of Mcm7, whereas the second pathway acts during S-phase of the following cell cycle and is mediated by Pif1-family helicases and especially by Rrm3.

## Discussion

When two replication forks converge during DNA replication termination in budding yeast cells (Fig. 9B, steps (ii) to (iii)), the Rrm3 and Pif1 DNA helicases help to unwind the final stretch of parental DNA between the two replisomes (Claussin et al, 2022; Deegan et al, 2019), thereby facilitating the bypass of the two CMG helicases (Fig. 9B, step (iv)). Rrm3 is the major player in this process in budding yeast cells (Deegan et al, 2019), analogous to the dominant role of Rrm3 in the disassembly of old CMG complexes from the previous cell cycle (Figs. 6–8 and EV5).

At a pair of converging replisomes, Rrm3-Pif1 migrate along the same parental strand as CMG from the opposing replisome (Fig. 9B), leading to a potential clash during DNA replication termination between Rrm3-Pif1 from one fork and CMG from the other. Nevertheless, Rrm3-Pif1 do not drive CMG disassembly under such conditions (Fig. 9B, step (iv)), and disintegration of the helicase is instead dependent upon SCF^Dia2 and the Cdc48 unfoldase (Maric et al, 2014). In contrast, our data suggest that old CMG helicases from the previous cell cycle are disassembled by Rrm3-Pif1 during the subsequent S-phase (Fig. 9C, steps (ii) to (iii)),

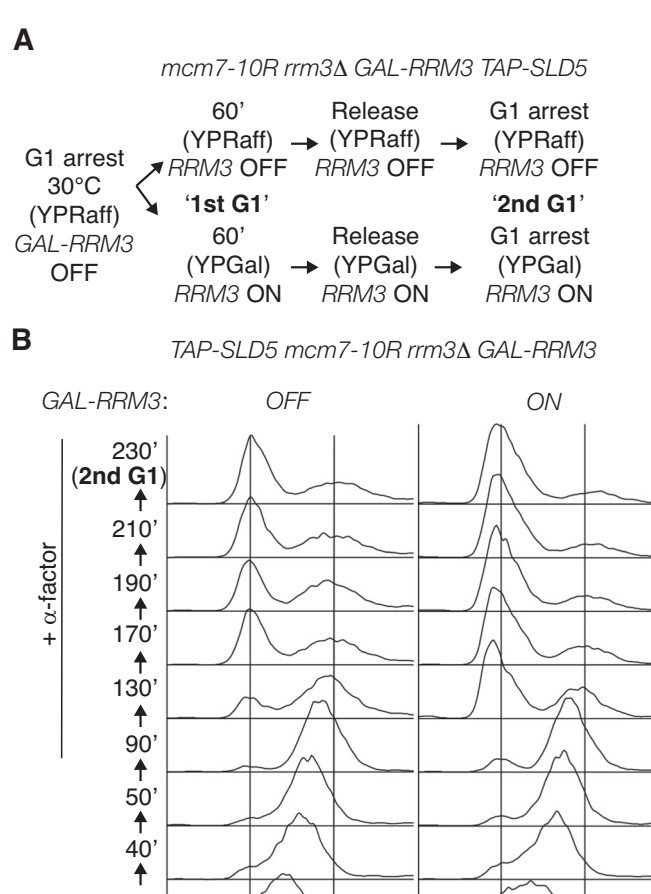

**A**

*mcm7-10R rrm3Δ GAL-RRM3 TAP-SLD5*

**B**

*TAP-SLD5 mcm7-10R rrm3Δ GAL-RRM3*

◀ **Figure 7.  Rrm3 promotes the disassembly of old CMG complexes after the G1 phase.**

(**A**) Reaction scheme involving *mcm7-10R rrm3Δ GAL-RRM3 TAP-SLD5* budding yeast cells (YCPR334). (**B**) DNA content from the experiment in (**A**) was monitored by flow cytometry. Upon release from the first arrest in G1 phase, mating pheromone (α-factor) was added again from 50 min onwards, to ensure that cells arrested subsequently in G1 phase of the second cell cycle. (**C**) Cells arrested in G1-phase arrest during the first and second cell cycles of the experiment in (**B**) were used to prepare cell extracts. The presence of CMG was monitored by immunoprecipitation of TAP-Sld5 and immunoblotting. Source data are available online for this figure.

**C**

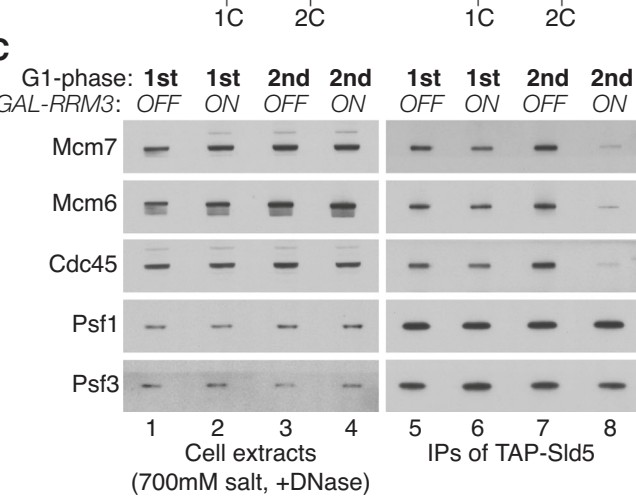

of new DNA replication forks with old CMG complexes increases the risk of DNA damage and is a major source of genome instability following defects in CMG helicase disassembly during DNA replication termination. The Rrm3-Pif1 pathway ameliorates the impact of old CMG helicase complexes on genome stability and is essential for cell viability when CMG ubiquitylation is defective in budding yeast cells. In the absence of both Rrm3 and Pif1, it is likely that old CMG complexes form a potent block to the progression or convergence of new DNA replication forks in the subsequent cell cycle.

Our data indicate that the Rrm3-Pif1 pathway disrupts old CMG complexes into their component parts (Figs. 5–7), rather than displacing intact helicase complexes from the DNA template. The disassembly of old CMG complexes by Rrm3 cannot occur during G1 phase but instead requires entry into S-phase and progression through the cell cycle (Fig. 7). This is consistent with the fact that Pif1-family helicases are stimulated by replication fork DNA (Lahaye et al, 1993) and can only displace MCM2–7 double hexamers in the context of a DNA replication fork (Hill et al, 2020). To analyse specifically the behaviour of old CMG complexes in *dia2Δ* cells that lack the ubiquitylation pathway for CMG disassembly, we previously expressed tagged GINS components or Cdc45 during G1 phase and found that CMG complexes assembled during one round of S-phase persisted until the end of G1 phase in the following cell cycle, despite the presence of Rrm3 and Pif1 (Maric et al, 2014). These data indicate that Rrm3-Pif1 can only act in the second S-phase to disassemble old CMG complexes (Fig. 9C). Moreover, recent work has found that Rrm3 function is dependent upon its interaction at replication forks with the CMG helicase and DNA polymerase ε (Olson et al, 2024). Mutations in Rrm3 that break these interactions severely impair the growth of *dia2Δ* cells, resembling the previously observed synthetic lethality of *rrm3Δ* with *dia2Δ*, and likely reflecting a near-lethal defect in CMG helicase disassembly. These findings support the idea that Rrm3 functions at DNA replication forks to disassemble old CMG.

Defining the mechanism by which Rrm3-Pif1 drive CMG disassembly is likely to require biochemical reconstitution of the encounter between a replication fork and old CMG. At this stage, multiple possibilities could be envisaged. For example, the disassembly of old CMG complexes might be dependent on a head-on collision between a replication fork and an old CMG complex that encircles dsDNA and translocates towards the fork. Alternatively, two replication forks might need to converge on an old CMG complex, before the latter can be disrupted by Rrm3-Pif1 (Fig. 9C). Such scenarios would contrast with the normal convergence of two replication forks during DNA replication

thereby allowing forks to converge and DNA synthesis to be terminated (Fig. 9C, steps (iv) to (v)).

The *mcm7-10R* allele delays CMG helicase disassembly and increases the proportion of cells with sub-nuclear foci of Rad52 in the following cell cycle (Fig. 5G). This suggests that the encounter

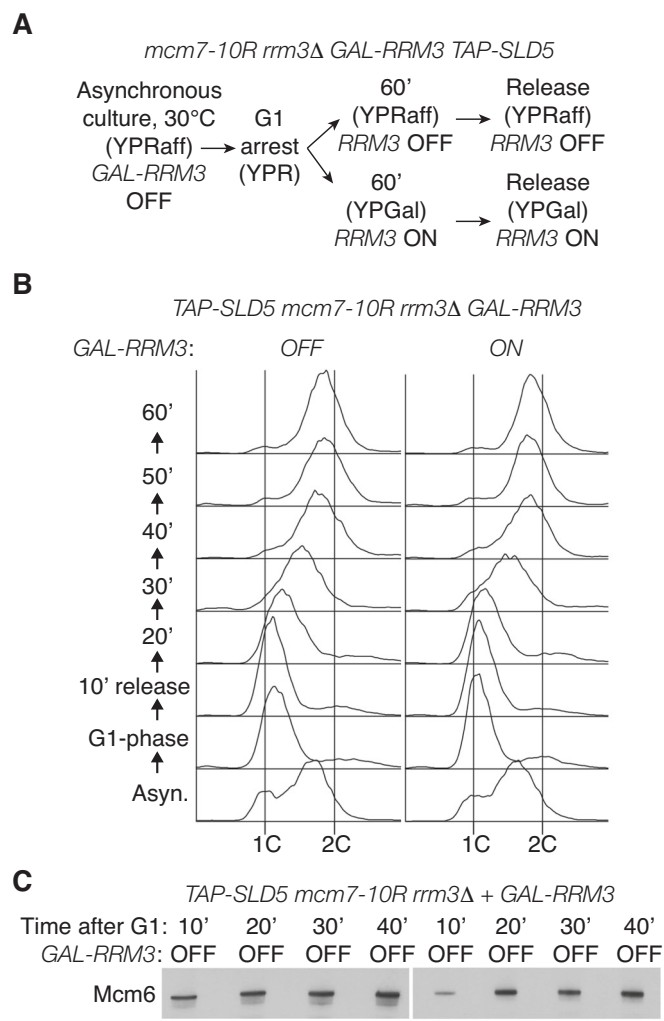

**A**

*mcm7-10R rrm3Δ GAL-RRM3 TAP-SLD5*

Asynchronous
culture, 30°C G1
(YPRaff) → arrest
*GAL-RRM3* (YPR)
OFF

60'
(YPRaff) → Release
*RRM3* OFF (YPRaff)
 *RRM3* OFF

60'
(YPGal) → Release
*RRM3* ON (YPGal)
 *RRM3* ON

**B**

*TAP-SLD5 mcm7-10R rrm3Δ GAL-RRM3*

*GAL-RRM3:* OFF ON

60'
50'
40'
30'
20'
10' release
G1-phase
Asyn.

1C 2C 1C 2C

**C**

*TAP-SLD5 mcm7-10R rrm3Δ + GAL-RRM3*

| Time after G1: | 10' | 20' | 30' | 40' | 10' | 20' | 30' | 40' |
|---|---|---|---|---|---|---|---|---|
| *GAL-RRM3:* | OFF | OFF | OFF | OFF | OFF | OFF | OFF | OFF |
| Mcm6 | | | | | | | | |
| Cdc45 | | | | | | | | |
| Psf1 | | | | | | | | |
| | 1 | 2 | 3 | 4 | 5 | 6 | 7 | 8 |

Cell extracts IPs of TAP-Sld5
(700mM KOAc, +DNase)

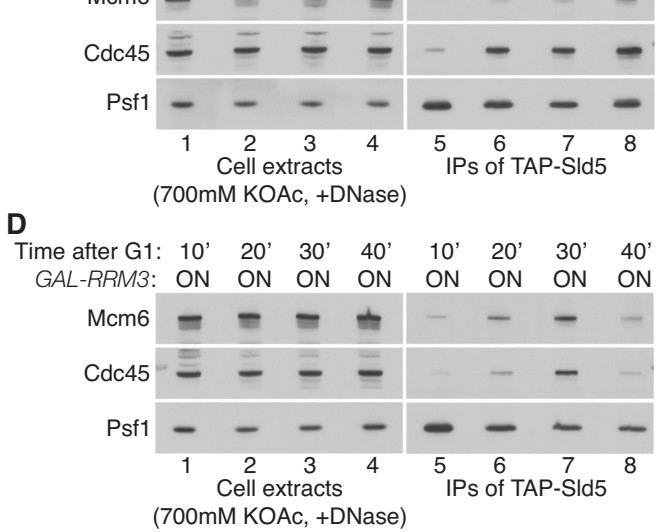

**D**

| Time after G1: | 10' | 20' | 30' | 40' | 10' | 20' | 30' | 40' |
|---|---|---|---|---|---|---|---|---|
| *GAL-RRM3:* | ON | ON | ON | ON | ON | ON | ON | ON |
| Mcm6 | | | | | | | | |
| Cdc45 | | | | | | | | |
| Psf1 | | | | | | | | |
| | 1 | 2 | 3 | 4 | 5 | 6 | 7 | 8 |

Cell extracts IPs of TAP-Sld5
(700mM KOAc, +DNase)

**Figure 8. Rrm3 acts during the S-phase to promote the disassembly of old CMG complexes.**

(A) Reaction scheme involving *mcm7-10R rrm3Δ GAL-RRM3 TAP-SLD5* budding yeast cells (YCPR334). (B) DNA content from the experiment in (A) was monitored by flow cytometry. (C) Samples of cells lacking Rrm3 (*GAL-RRM3* OFF) from the experiment in (B) were taken at the indicated times after release from G1 arrest and then used to prepare cell extracts. CMG was monitored by immunoprecipitation of TAP-Sld5 and immunoblotting. (D) Equivalent analysis for cells in which Rrm3 induction was induced after arresting cells in G1 phase (*GAL-RRM3* ON). Source data are available online for this figure.

to be disassembled via ubiquitylation and Cdc48-Ufd1-Npl4 (Fig. 9B). It remains to be determined how the mechanism of CMG disassembly by Rrm3-Pif1 compares with the displacement of Mcm2–7 double hexamers during elongation. In vitro studies indicate that Pif1 displaces Mcm2–7 when a distal nucleosome or a converging replisome provides resistance and prevents Pif1 from simply pushing the double hexamers along dsDNA (Hill et al, 2020).

A previous study of human CUL2[LRR1] suggested that blocking CMG helicase disassembly during DNA replication termination in early replicons can interfere with CMG assembly at later origins, by preventing the recycling of CMG components such as CDC45 that might be present at limiting levels in some human cells (Fan et al, 2021). However, inactivation of CUL2[LRR1] in *Xenopus* egg extracts does not affect the progression of DNA replication (Dewar et al, 2017; Sonneville et al, 2017), indicating that the sequestering on chromatin of limiting replisome components might only be an issue in specific cell types. In budding yeast, the efficiency of CMG assembly is similar in control cells and in cells lacking Dia2 (Maric et al, 2014), indicating that the sequestering of replisome components by old CMG complexes is unlikely to contribute to the genome instability that results from defects in helicase disassembly.

The existence of the Rrm3-Pif1 pathway has important implications for the evolution of CMG helicase disassembly in eukaryotic cells. The fact that fungi and animals ubiquitylate the Mcm7 subunit of CMG during DNA replication termination, using evolutionarily distinct ubiquitin ligases that are not conserved in plants, suggests that CMG ubiquitylation might have arisen multiple times during eukaryotic evolution. Therefore, ancestral eukaryotes must have disassembled the CMG helicase by alternative means that predated the ubiquitylation of Mcm7. Pif1-family helicases are broadly conserved in eukaryotic species and could have mediated such a pathway, acting during the subsequent S-phase to disassemble old CMG complexes from the previous cell cycle. The phenotypes of *mcm7-10R* indicate that delaying CMG to the subsequent cell cycle in ancestral eukaryotes would have come at the price of higher genome instability, thereby providing a selective pressure for the repeated emergence of CMG ubiquitylation, which then provided a mechanism for the rapid disassembly of CMG complexes during DNA replication termination in a single cell cycle.

Given the diversity of regulation between fungi and animal cells, it would be interesting in future studies to explore how CMG helicase disassembly is controlled in plants and in other eukaryotes for which chromosome duplication remains poorly characterised. It also remains to be determined whether Pif1-family helicases

termination, when Rrm3-Pif1 at one fork cannot disassemble CMG from the opposing fork (Fig. 9B). It is possible that Rrm3-Pif1 are released from the lagging strand DNA template under such conditions, once the final stretch of parental DNA between the converged forks has been unwound, leaving the two CMG helicases

## A

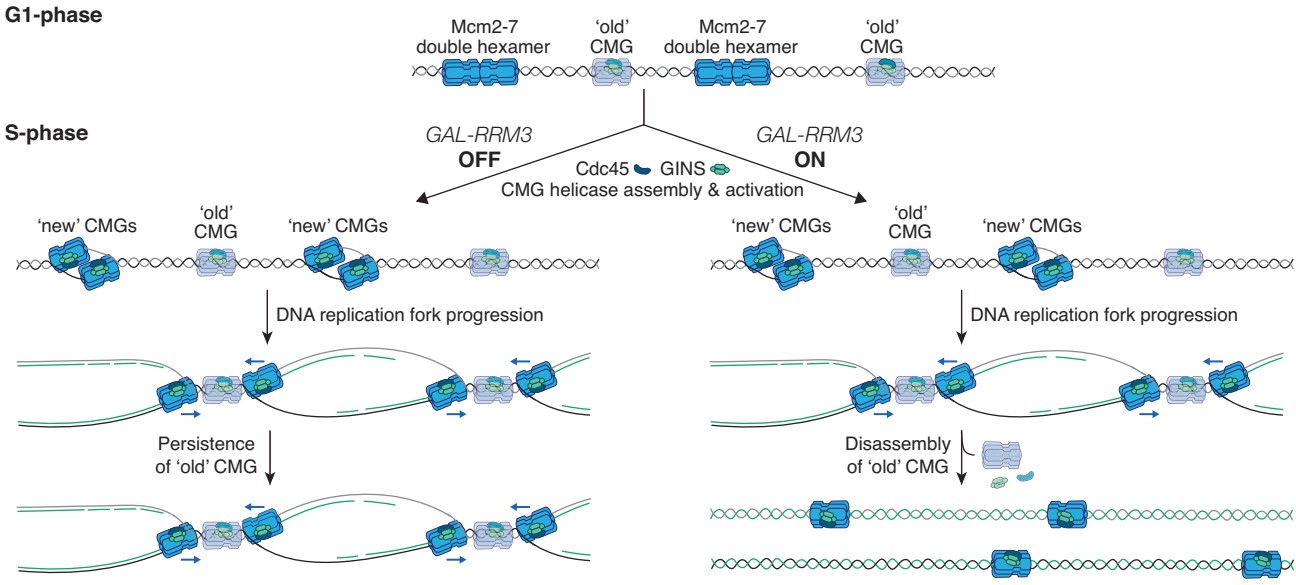

*mcm7-10R rrm3Δ GAL-RRM3* cells (*GAL-RRM3* **OFF**)

**G1-phase**

Mcm2-7 double hexamer · 'old' CMG · Mcm2-7 double hexamer · 'old' CMG

**S-phase**

*GAL-RRM3* **OFF** ← → *GAL-RRM3* **ON**

Cdc45 · GINS
CMG helicase assembly & activation

'new' CMGs · 'old' CMG · 'new' CMGs

DNA replication fork progression

Persistence of 'old' CMG

Pif1 likely contributes to eventual disassembly of 'old' CMG complexes from previous cell cycle.

'New' CMG complexes from this cell cycle persist on chromosomes for a period after DNA replication termination, due to ubiquitylation defect of *mcm7-10R*.

'old' CMG · 'new' CMGs

DNA replication fork progression

Disassembly of 'old' CMG

'New' CMG complexes from this cell cycle persist on chromosomes for a period after DNA replication termination, due to ubiquitylation defect of *mcm7-10R*.

## B

Wild type cells

**S-phase**

(i)

DNA replication fork progression

(ii) Rrm3/Pif1 · Rrm3/Pif1 · Rrm3/Pif1 · Rrm3/Pif1

Fork convergence: Rrm3 & Pif1 help unwind parental DNA

(iii) Rrm3/Pif1 · Rrm3/Pif1

CMG bypass; transition of CMG to dsDNA

(iv)

Mcm2-6 · Mcm7-Ubi · SCF^Dia2 + Cdc48 · CMG ubiquitylation & disassembly

**G2/M-phase**

## C

*dia2Δ* cells (or *mcm7-10R* with short G1-phase)

**S-phase**

(i) 'old' CMG · 'old' CMG

DNA replication fork progression

(ii) Rrm3/Pif1 · N · C · Rrm3/Pif1 · N · C · C · N · Rrm3/Pif1 · C · N · Rrm3/Pif1

Disassembly of 'old' CMG by Rrm3 / Pif1

(iii) Rrm3/Pif1 · Rrm3/Pif1 · Rrm3/Pif1 · Rrm3/Pif1

Fork convergence: Rrm3 & Pif1 help unwind parental DNA

(iv) Rrm3/Pif1 · Rrm3/Pif1

CMG bypass; transition of CMG to dsDNA

(v)

CMG persists on chromosomes after termination

**G2/M-phase**

**Figure 9. Models for disassembly of old CMG complexes by Rrm3-Pif1.**

(A) Scheme illustrating the behaviour of old and new CMG complexes for the experiment in Fig. 8. (B) During DNA replication termination in wild-type budding yeast cells, Rrm3 and Pif1 help to unwind the last stretch of parental DNA between converging forks during DNA replication termination, but do not trigger CMG helicase disassembly. See text for details. 'N' (amino terminal) and 'C' (carboxy terminal) refer to the orientation of the Mcm2–7 ring in the CMG helicase. (C) *dia2Δ* cells (or *mcm7-10R* cells with a short G1 phase) enter S-phase with old CMG complexes still present on chromatin. These old CMG complexes are disassembled during S-phase, dependent upon Rrm3 (likely with backup from Pif1—see text for further details). Subsequently, Rrm3-Pif1 helps to unwind the last stretch of parental DNA between converging forks during DNA replication termination, leaving the new CMG complexes from the current cell cycle on chromatin, due to the defect in CMG ubiquitylation.

mediate CMG helicase disassembly in other species apart from budding yeast, and whether this role is shared with other 5′ to 3′ DNA helicases that help replication forks to bypass roadblocks, such as the RTEL1 helicase in metazoa (Hourvitz et al, 2023; Vannier et al, 2014).

# Methods

Reagents and resources from this study are listed in Appendix Table S3 and are available from MRC PPU Reagents and Services (https://mrcppureagents.dundee.ac.uk) or upon request.

## Yeast strains and growth

Yeast strains employed in this study are based on the W303 background and are listed in Appendix Table S3. Cells were grown in Yeast Extract Peptone (YP) medium, consisting of 1% (w/v) Yeast extract and 2% (w/v) Peptone. The medium was further supplemented with either 2% glucose (YPD medium), 2% raffinose (YPR medium), or 2% galactose (YPG medium). Unless specified otherwise, yeast cultures were grown at 30 °C.

To synchronize *MAT***a** haploid cells in the G1 phase of the cell cycle, mid-log cell cultures were treated with mating pheromone (α-factor) at a concentration of 7.5 µg/mL. After 1 h, additional aliquots of 2.5 µg/mL α-factor (or 5 µg/mL in the case of *dia2Δ* cells) were added every 15 min until more than 90% of cells were unbudded or exhibited 'shmoos'. To release from the G1 arrest into S-phase, cells were washed and resuspended in fresh media lacking the pheromone and then left to grow.

To arrest cells in early S-phase, cells were released from G1 arrest and transferred into fresh media supplemented with 0.2 M hydroxyurea until ~90% of the cells had initiated bud formation.

Degradation of Cdc48-AID was induced by the addition of 0.5 mM of the auxin 3-indoleacetic acid to the cell culture.

## Meiotic progeny analysis of diploid cells

Diploid yeast cells were induced to enter meiosis by culturing them in Rich Sporulation Media (RSM) containing 0.25% (w/v) yeast extract, 1.5% (w/v) potassium acetate, 0.1% (w/v) glucose, 2% (w/v) agar and 770 mg/L 'supplement mixture' comprising 100 mg/L each of adenine and uracil, 50 mg/L each of L-histidine, L-leucine, L-methionine, L-arginine, L-lysine and L-tryptophan, 20 mg/L L-Tyrosine and 250 mg/L of L-phenylalanine. Ascus walls were digested for 20 min with β-glucuronidase and spores were then separated using a dissection microscope (Singer Instruments). After growth on a YPD plate for 2 days at 30 °C, the genotype of each colony was checked by replica plating onto selective media.

## Analysis of yeast cell growth by serial dilution on solid medium

A single colony from each yeast strain was suspended in 1 mL of 1× PBS at pH 7.4. Subsequently, cells were counted and adjusted to a final cell density of $0.33 \times 10^7$ cells/mL, and three consecutive tenfold dilutions were prepared and vortexed.

Finally, 15 µL drops of each dilution were spotted on the plates (the location of each drop being determined by a grid placed below the plate), resulting in a total of $5 \times 10^4$, $5 \times 10^3$, $5 \times 10^2$ and 50 cells. After incubation and colony formation, the plates were imaged using an Epson Expression 10000XL scanner every 24 h for up to 5 days, depending on the conditions.

## Flow cytometry analysis

To monitor DNA content, a 1 mL aliquot of cells (~$10^7$ cells) was fixed by resuspending them in 70% (v/v) ethanol. Following fixation, cells were processed as described previously (Labib et al, 1999) and analysed with a FACSCanto II flow cytometer (Becton Dickinson), and FlowJo software. Samples were gated manually after data processing to exclude cell fragments and other particles.

## Antibody coupling to magnetic beads

To initiate the antibody coupling process, 425 µL of magnetic Dynabeads M-270 epoxy beads (Thermo Scientific, cat. No. 14302D) were placed in a magnetic rack, and any residual DMF storing solution was carefully removed from the beads. Subsequently, the beads were thoroughly washed with 1 mL of 1 M sodium phosphate pH 7.4 with agitation on a rotating wheel for 10 min. Following this, the supernatant was discarded and the beads were additionally washed twice with sodium phosphate.

For antibody coupling, 320 µg of anti-Sld5 antibody was combined with 300 µL of 3 M ammonium sulphate and the appropriate amount of 1 M sodium phosphate pH 7.4, to give a total volume of 900 µL. The mixture was then incubated for 2 days at 4 °C with constant agitation.

Beads were then placed in a magnetic rack and the supernatant was removed. The beads were then washed four times with 1× PBS before a 10 min incubation in PBS/0.5% NP-40. Following this step, the beads were incubated with PBS/BSA twice for 5 min, and finally resuspended in 900 µL of PBS/BSA solution supplemented with 0.02% sodium azide.

## Immunoprecipitation of protein complexes from yeast cell extracts

CMG complexes were isolated from yeast cell cultures as described previously (De Piccoli et al, 2012; Gambus et al, 2006; Maric et al,

2014). Briefly, yeast cell pellets from a 250 mL culture, were centrifuged at 200 × g for 3 min. The initial pellet was washed with 50 mL of Tris-Acetate pH 9, followed by a second wash with 10 mL of lysis buffer (100 mM Tris-Acetate pH 9, 700 mM potassium acetate, 10 mM magnesium acetate, 2 mM EDTA). Cells were then pelleted, weighed, and suspended in three volumes of lysis buffer supplemented with 2 mM sodium fluoride, 2 mM sodium β-glycerophosphate pentahydrate, 1 mM dithiothreitol (DTT), 1% Protease Inhibitor Cocktail (P8215, Sigma-Aldrich), and 1× Complete Protease Inhibitor Cocktail (05056489001, Roche).

The suspended cells were frozen dropwise in a 50 mL Falcon tube filled with liquid nitrogen, resulting in the formation of 'yeast popcorn'. After allowing the liquid nitrogen to evaporate, the popcorn was stored at −80 °C. Subsequently, equal amounts of popcorn for each sample were ground in a SPEX SamplePrep 6780 Freezer/Mill, using two cycles at 'rate 14' (each cycle consisting of 2 min of 'run' and 2 min of 'cool'). The resulting yeast cell powder was recovered and allowed to thaw at room temperature. Cell extracts were then transferred to centrifuge tubes on ice, with volume measurements taken along the way. Each extract was then diluted with a 1/4 volume of lysis buffer supplemented with 50% glycerol, 700 mM potassium acetate, 10 mM magnesium acetate, 0.5% IGEPAL CA-630, 2 mM EDTA, inhibitors, and DTT at the concentrations mentioned above. Subsequently, 800 U/mL of DNase (Pierce Universal Nuclease) was added to each extract, followed by a 30-min incubation at 4 °C on a rotation wheel to release CMG complexes from chromatin. Insoluble cell debris was then removed by centrifugation at 25,000 × g for 30 min, followed by ultracentrifugation at 100,000 × g for 1 h.

From each of the resulting extracts, a 50 μL aliquot was mixed with 100 μL 1.5× Laemmli Buffer and heated at 95 °C for 2 min. The remainder of each extract was combined with 100 μL antibody-coupled magnetic beads (~1.7 × 10⁹ beads), ensuring that a fixed volume of extract was used for every sample in each experiment. Immunoprecipitation of protein complexes occurred over a 2-h period at 4 °C on a rotation wheel. Following incubation, proteins bound to magnetic beads were washed four times with 1 mL of IP Wash Buffer (100 mM Tris-Acetate pH 9, 100 mM potassium acetate, 10 mM magnesium acetate, 2 mM EDTA, 0.1% IGEPAL CA-630). Eluted protein complexes were then separated from beads by adding 50 μL of 1× Laemmli Buffer and heating the suspension at 95 °C for 5 min. The supernatant was then removed and stored at −80 °C until immunoblotting analysis.

## Mass spectrometry analysis

Samples were purified from yeast cell extracts synchronized in mid S-phase as above and eluted in 40 μl Laemmli buffer, of which 35 μl was resolved by SDS-polyacrylamide gel electrophoresis (SDS-PAGE) using NuPAGE Novex 4-12% Midi Bis-Tris gels (NP0321, Life Technologies) with NuPAGE MOPS SDS buffer (NP000102, Life Technologies). Subsequently, gels were stained with 'Simply-Blue SafeStrain' colloidal Coomassie (LC6060, Invitrogen) following the manufacturer's instructions, and each lane was cut into 40 slices that were digested with trypsin before processing for mass spectrometry (MS Bioworks, USA). Data were analysed using Scaffold software (Proteome Software Inc, USA). The mass spectrometry data for the experiments in Datasets EV1 and EV2 have been deposited to the ProteomeExchange Consortium via the

PRIDE partner repository database (Perez-Riverol et al, 2022) with the dataset identifier PXD048935 and 10.6019/PXD048935.

## Purification of recombinant CMG helicase complexes

Recombinant CMG helicase complexes were generated in budding yeast cells by co-expression of the 11 subunits from the bidirectional GAL1,10 promoter. Each coding sequence was codon optimised to facilitate high-level expression, as described previously (Frigola et al, 2013), and the yeast Gal4 transcription factor was also co-expressed to enhance expression from the GAL1,10 promoter. Cells with integrated plasmids expressing the 11 subunits of a particular version of CMG were cultured at 30 °C in YPR media until reaching a density of 2–3 × 10⁷ cells/mL. Expression from the GAL1,10 promoter was then induced by adding 2% galactose to the media, and cultures were allowed to grow at 30 °C for 3 h. Typically, 12 l of cells were utilized for each CMG purification.

Cells were harvested by centrifugation at 1935 × g for 10 min, and the pellets were washed once with CMG Wash Buffer (25 mM HEPES-KOH pH 7.6, 10% (v/v) glycerol, 300 mM KCl, 2 mM MgOAc, 0.02% (v/v) Tween-20, 1 mM DTT), before resuspension in 0.3 volumes (relative to pellet mass) of CMG Buffer (same composition as CMG Wash Buffer but including 1× Complete Protease inhibitor cocktail). Cell suspensions were frozen dropwise in a 500 mL container filled with liquid nitrogen, resulting 'yeast popcorn', as described above. This material was ground to a fine powder using a SPEX CertiPrep 6850 FreezerMill for four cycles of 2 min, each at a rate of 15. Each cycle consisted of 2 min of 'run' and 2 min of 'cool,' and the resulting powder was stored at −80 °C.

Subsequently, the frozen powder was thawed at room temperature and mixed with one volume of CMG Buffer. The mixture was then centrifuged at 235,000 × g for 1 h at 4 °C. The soluble extract, typically around 40 mL, was recovered and mixed with 4 mL of anti-FLAG M2 affinity resin for 3 h at 4 °C on a rotating wheel. Subsequently, the resin was collected and extensively washed (typically in 20 column volumes) with CMG Buffer, and an additional column volume with CMG Wash Buffer. CMG complexes were then eluted by incubating the beads for 30 min on ice in 1 column volume of CMG FLAG Elution Buffer (same composition as CMG Wash Buffer supplemented with 0.5 mg/mL FLAG peptide), and the eluted material then separated from the beads. This step was repeated by incubating the beads with CMG FLAG Elution Buffer that contained 0.25 mg/mL FLAG peptide, and both eluted samples were pooled in the same tube.

FLAG eluates were then adjusted to 2 mM CaCl₂ and mixed with 3 mL of Calmodulin affinity resin before incubation for 1 h at 4 °C on a rotating wheel. The beads were then collected and mixed with 10 mL of CMG Calmodulin Elution Buffer (25 mM HEPES-KOH pH 7.6 10% (v/v) glycerol, 300 mM KCl, 2 mM MgOAc, 0.02% (v/v), Tween-20, 2 mM EDTA, 2 mM EGTA and 1 mM DTT). Subsequently, the eluates were collected and pooled.

The eluate fraction was loaded onto a 0.2 mL MiniQ column (GE Healthcare Lifesciences) in CMG Wash Buffer. CMG complexes were eluted from the MiniQ column using a 4 mL gradient ranging from 0.3 M to 0.6 M KCl, which was automatically mixed by an Akta system. After elution, fractions were analysed by SDS-PAGE followed by Coomassie staining, and those containing CMG were pooled and dialysed overnight at 4 °C against CMG Dialysis Buffer using a Slide-A-Lyzer dialysis cassette

(Thermo). The dialysed sample was recovered, aliquoted, and snap-frozen. The final concentration of CMG complexes was determined by SDS-PAGE followed by Coomassie staining, comparing it with a titration of known concentrations of Bovine Serum Albumin (BSA).

## In vitro CMG ubiquitylation assay

To assemble reconstituted reactions for the CMG ubiquitylation process, ubiquitylation enzymes were purified using previously established protocols (Deegan et al, 2019; Deegan et al, 2020; Yeeles et al, 2015).

The reactions, typically 8 µL in volume, comprised 15 nM CMG, 30 nM Uba1 (E1 enzyme, provided by Axel Knebel, MRC PPU), 15 nM Cdc34 (E2 enzyme), 1 nM or 25 nM SCF$^{Dia2}$ (E3 enzyme), 30 nM Ctf4, 30 nM Pol ε, 45 nM Mrc1, 6 µM Ubiquitin (depending on the experiment, either wild-type ubiquitin, or lysine-free ubiquitin known as K0-Ubiquitin) and 5 mM ATP. Reactions were assembled on ice in Reaction Buffer (25 mM HEPES-KOH pH 7.6, 75 mM KOAc, 10 mM MgOAc, 0.02% IGEPAL CA-630, 0.1 mg/mL BSA, 1 mM DTT) and then incubated for 20 min at 30 °C, unless stated otherwise. To halt the reactions, 2X LDS Buffer (Thermo-Fisher Scientific) was added, and the mixture was boiled at 95 °C for 5 min.

For the experiment in Fig. EV3B, ubiquitylation reactions were performed as mentioned above, and each sample was incubated at 4 °C with 2.5 µL of magnetic beads coupled to anti-Cdc45 antibody. Protein mixtures were incubated for 1 h and CMG complexes bound to the beads were washed twice with 190 µL of Reaction Buffer containing 150 mM KOAc. Finally, beads were resuspended in 1× LDS and boiled for 5 min at 95 °C.

In experiments involving Mcm7 with inserted TEV cleavage sites, CMG ubiquitylation reactions were performed in Reaction Buffer containing NaOAc instead of KOAc, and then stopped by adding 700 mM NaOAc to the reaction mix. Subsequently, the mixture was incubated with 0.5 mg/mL TEV protease (provided by Axel Knebel, MRC PPU) for 40 min at 30 °C.

## In vitro CMG disassembly assay

The Cdc48 segregase and the Ufd1-Npl4 heterodimer were purified as described previously (Deegan et al, 2020). Following ubiquitylation reactions as above, 50 nM Cdc48 and 50 nM Ufd1-Npl4 were added and incubation continued for an additional 20 min at 30 °C. Subsequently, 700 mM NaOAc was added to stop the reaction before the addition of 2.5 µL of magnetic beads coupled with an antibody recognizing the Sld5 subunit of the CMG helicase. The mixture was incubated for 40 min at 4 °C with constant shaking at 1400 rpm on an Eppendorf Thermomixer. Subsequently, bound proteins were washed twice with Reaction Wash buffer (25 mM HEPES-KOH pH 7.6 150 mM KOAc, 10 mM MgOAc, 0.02% (v/v) IGEPAL CA-630, 0.1 mg/mL BSA and 1 mM DTT), and then eluted by addition of 2× LDS Buffer and heating at 95 °C for 5 min.

## In vitro assay to monitor SCF$^{Dia2}$ interaction with CMG

To assess the association of SCF$^{Dia2}$ with CMG complexes, 20 µL reactions were prepared as for the ubiquitylation reactions described above, except that the SCF$^{Dia2}$ concentration was increased to 10 nM to facilitate detection by immunoblotting, and ubiquitin and ATP were excluded. Reactions were incubated for 15 min on ice and input samples (5 µL) then collected. Subsequently, 3.75 µL of magnetic beads coupled to anti-Sld5 antibodies were added to the remainder of each reaction, before a 30-min incubation at 4 °C with constant shaking at 1,400 rpm on an Eppendorf Thermomixer. Bound CMG complexes were recovered, washed twice in Reaction Wash buffer and then eluted by adding 2X LDS Buffer and heating at 95 °C for 5 min.

## SDS-PAGE and immunoblotting

Proteins were separated on NuPAGE Novex 4–12% Bis-Tris gels with NuPAGE MOPS SDS buffer. For better resolution of ubiquitin chains, NuPAGE 3–8% Tris-Acetate gels were used in combination with NuPAGE Tris-Acetate SDS buffer. Subsequently, proteins were either stained with colloidal Coomassie blue dye or transferred to a nitrocellulose membrane utilizing the iBlot Dry Transfer System (Life Technologies) in accordance with the manufacturer's instructions.

The antibodies used to detect the proteins in this study are listed in Appendix Table S3.

## Detection of sub-nuclear Rad52-GFP foci in yeast cells

To detect Rad52-GFP foci, 1 mL of yeast cell cultures were harvested and fixed by mixing with 1 mL 16% paraformaldehyde solution (Thermo Scientific, cat. No. 043368), to give a final concentration of 8% Paraformaldehyde. The mixture was incubated for 10 min at room temperature on a rotating wheel and subsequently washed in 1× PBS. After the wash, cells were resuspended in 1 mL 1× PBS and stored at 4 °C. To visualize DNA, cells were pelleted at 845 g for 3 min and incubated for 30 min at room temperature in 500 µL of 1 µg/mL DAPI. Subsequently, cells were washed three times in 1× PBS to reduce background fluorescence and then finally resuspended in a suitable volume for microscopy (typically 20 µL). A 4 µL aliquot was then placed onto a microscope slide and mounted with a coverslip sealed with nail polish. Cells were analysed using a Yokogawa CSU-X1 spinning disk microscope with a HAMAMATSU C13440 camera, equipped with a PECON incubator and a 63×/1.40 Plan Apochromat oil immersion lens (Olympus). Multiple fields were imaged for each strain. Images were captured using ZEN blue software (Zeiss), processed, and analysed with ImageJ software (National Institute of Health).

Approximately 100 cells per strain were counted in three independent experiments, to determine the percentage with Rad52-GFP foci. Mean values were then calculated, together with the associated standard deviation. Samples were compared using Prism9 software (GraphPad) to assess statistical significance, as described in the figure legends. A one-way ANOVA was used for Figs. 5F and 6C, since the data involve a single independent variable (% cells with Rad52-GFP foci), whereas a two-way ANOVA was used for Fig. 5G, in which the data involve two independent variables (% cells with Rad52-GFP foci, and time after release from G1 arrest). In all cases, Tukey's test was then used after the ANOVA, to determine which group's means were statistically different.

## Sequence alignment and structural analysis

For Fig. EV2A, protein sequences were aligned with Clustal Omega software (https://www.ebi.ac.uk/Tools/msa/clustalo/) and the alignment displayed with MView (https://www.ebi.ac.uk/Tools/msa/mview/).

AlphaFold2-multimer (Mirdita et al, 2022) was used to generate the model in Fig. EV2D and Movie EV2 of SCF[Dia2]. Structural models were displayed, and Movies generated, using UCSF Chimera (Pettersen et al, 2021).

## Data availability

The mass spectrometry datasets that were produced in this study are available in the PRIDE repository database (https://www.ebi.ac.uk/pride/) with the dataset identifier PXD048935 and 10.6019/PXD048935.

The source data of this paper are collected in the following database record: biostudies:S-SCDT-10_1038-S44318-024-00161-x.

## Peer review information

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

## Acknowledgements

The authors thank Ryo Fujisawa for help with protein purification, MRC PPU Reagents and Services (https://mrcppureagents.dundee.ac.uk) for antibody production and Ottavia Olson for sharing unpublished data. We are grateful for the support of the Medical Research Council (core grants MC_UU_12016/13 to KL and MC_UU_0035/4 to TD) and Cancer Research UK (Programme Grant C578/A24558 and PhD studentship C578/A25669 to KL). Materials generated in this study are listed in Appendix Table S3 and are available from MRC PPU Reagents and Services (https://mrcppureagents.dundee.ac.uk) or upon request.

## Author contributions

**Cristian Polo Rivera**: Data curation; Formal analysis; Validation; Investigation; Visualization; Methodology; Writing—review and editing. **Tom D Deegan**: Conceptualization; Data curation; Formal analysis; Supervision; Methodology; Writing—review and editing. **Karim PM Labib**: Conceptualization; Data curation; Formal analysis; Supervision; Funding acquisition; Visualization; Methodology; Writing—original draft; Project administration.

Source data underlying figure panels in this paper may have individual authorship assigned. Where available, figure panel/source data authorship is listed in the following database record: biostudies:S-SCDT-10_1038-S44318-024-00161-x.

## Disclosure and competing interests statement

Karim Labib is a member of the Advisory Editorial Board of *The EMBO Journal*. This has no bearing on the editorial consideration of this article for publication.

# Expanded View Figures

**Figure EV1.   Model for disassembly of CMG helicase in wild-type cells and *dia2Δ*.**

(**A**) In wild-type budding yeast cells, ubiquitylation of CMG by SCF[Dia2] is sterically impeded at replication forks, by the parental DNA strand that is excluded from the Mcm2–7 ring of the helicase. This inhibition is released during DNA replication termination, when a pair of forks converge and the two CMG helicases bypass each other, thereby breaking the association between CMG and the excluded parental DNA strand. (**B**) In *dia2Δ* cells, CMG cannot be ubiquitylated during DNA replication termination and persists on chromatin until the next cell cycle. However, once *dia2Δ* enter S-phase, the old CMG complexes are disassembled by a previously unknown pathway, likely coupled to the encounter between new replication forks and old CMG.

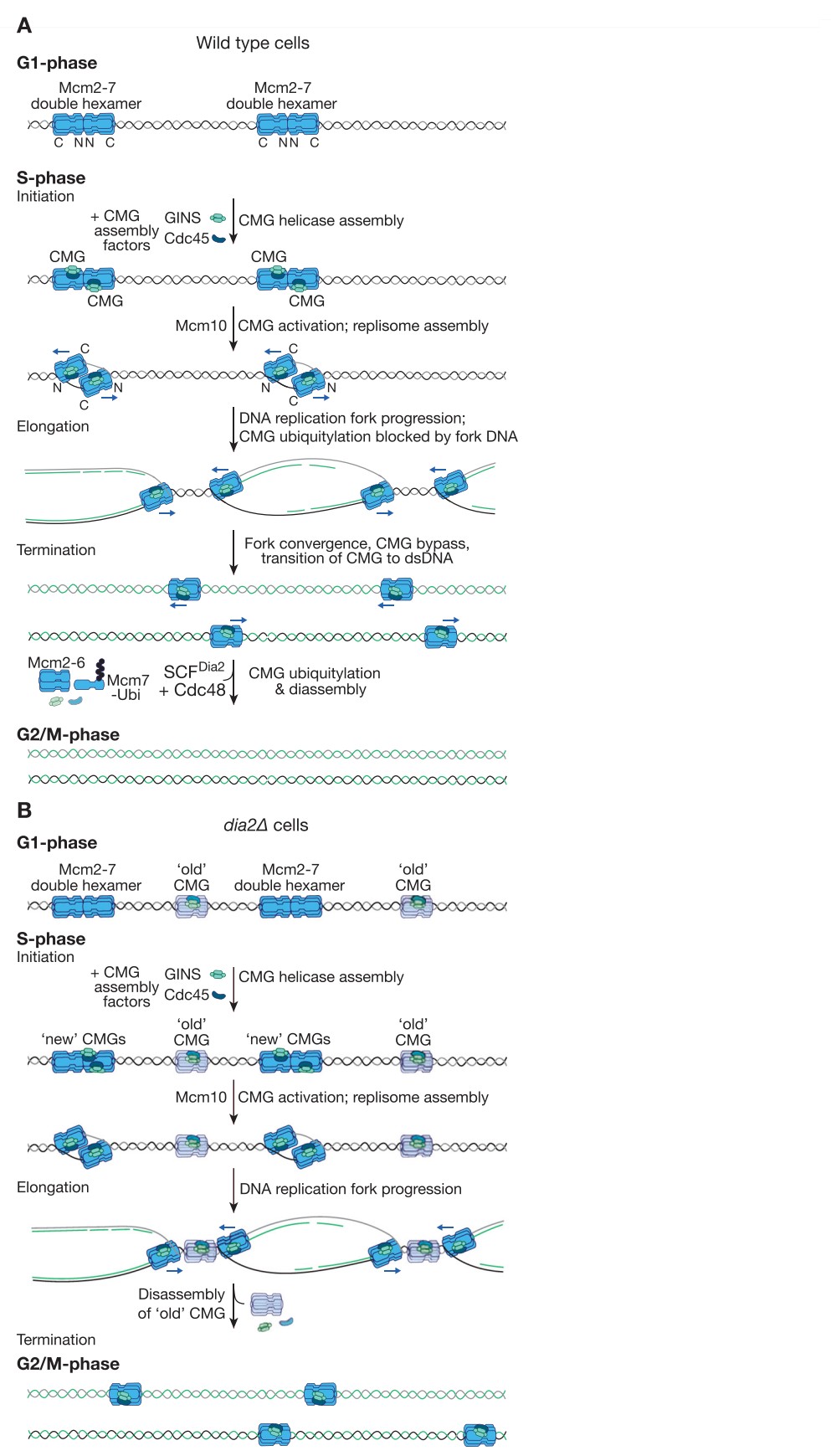

**A**

Wild type cells

**G1-phase**

Mcm2-7 double hexamer

Mcm2-7 double hexamer

**S-phase**

Initiation

+ CMG assembly factors · GINS · Cdc45 | CMG helicase assembly

CMG

CMG

CMG

CMG

Mcm10 | CMG activation; replisome assembly

Elongation

DNA replication fork progression; CMG ubiquitylation blocked by fork DNA

Termination

Fork convergence, CMG bypass, transition of CMG to dsDNA

Mcm2-6 — Mcm7-Ubi | SCF^Dia2 + Cdc48 | CMG ubiquitylation & diassembly

**G2/M-phase**

**B**

dia2Δ cells

**G1-phase**

Mcm2-7 double hexamer

'old' CMG

Mcm2-7 double hexamer

'old' CMG

**S-phase**

Initiation

+ CMG assembly factors · GINS · Cdc45 | CMG helicase assembly

'new' CMGs

'old' CMG

'new' CMGs

'old' CMG

Mcm10 | CMG activation; replisome assembly

Elongation

DNA replication fork progression

Disassembly of 'old' CMG

Termination

**G2/M-phase**

CMG complexes persist on chromosomes after DNA replication termination, until S-phase of the next cell cycle.

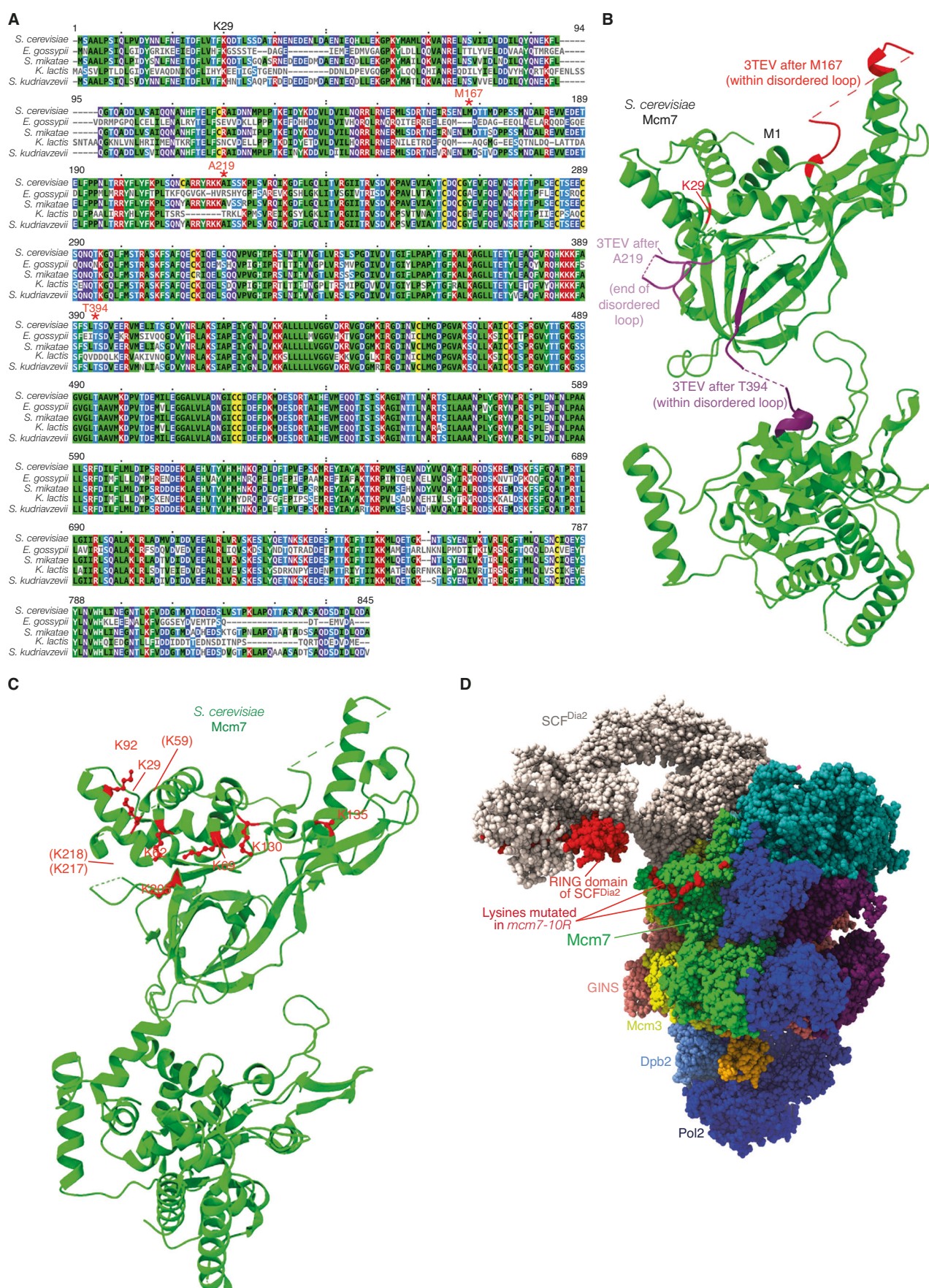

◀ **Figure EV2.   Location of TEV cleavage sites and SCF^Dia2 ubiquitylation sites in Mcm7.**

(A) Mcm7 from multiple yeast species (*S. cerevisiae* = *Saccharomyces cerevisiae*; *E. gossypii* = *Eremothecium gossypii*; *S. mikatae* = *Saccharomyces mikatae*; *K. lactis* = *Kluyveromyces lactis*; *S. kudriavzevii* = *Saccharomyces kudriavzevii*) were aligned using Clustal Omega software. Mcm7-K29 is marked in black, whereas the three sites used for insertion of TEV cleavage sites (M167, A219 and T394) are shown in red with an asterisk. (B) The structure of *S. cerevisiae* Mcm7 (from PDB file 7PMK), illustrating the location of K29 and the three sites within disordered loops that were used to insert three consecutive TEV cleavage sites (after M167, A219 or T394 of Mcm7). Also see Movie EV1. (C) Structure of yeast Mcm7 (from PDB file 7PMK) and location in red of the 10 lysines mutated in Mcm7-10R. Note that K217 and K218 are in a disordered loop not visible in the cryoEM structure of the yeast replisome. (D) The structure of SCF^Dia2 was predicted with AlphaFold2-Multimer and then docked onto the cryoEM structure of the yeast replisome (adapted from PDB file 7PMK) with UCSF Chimera software. The RING domain of Dia2, and the 10 lysines mutated in Mcm7-10R, are shown in red. Also see Movie EV2.

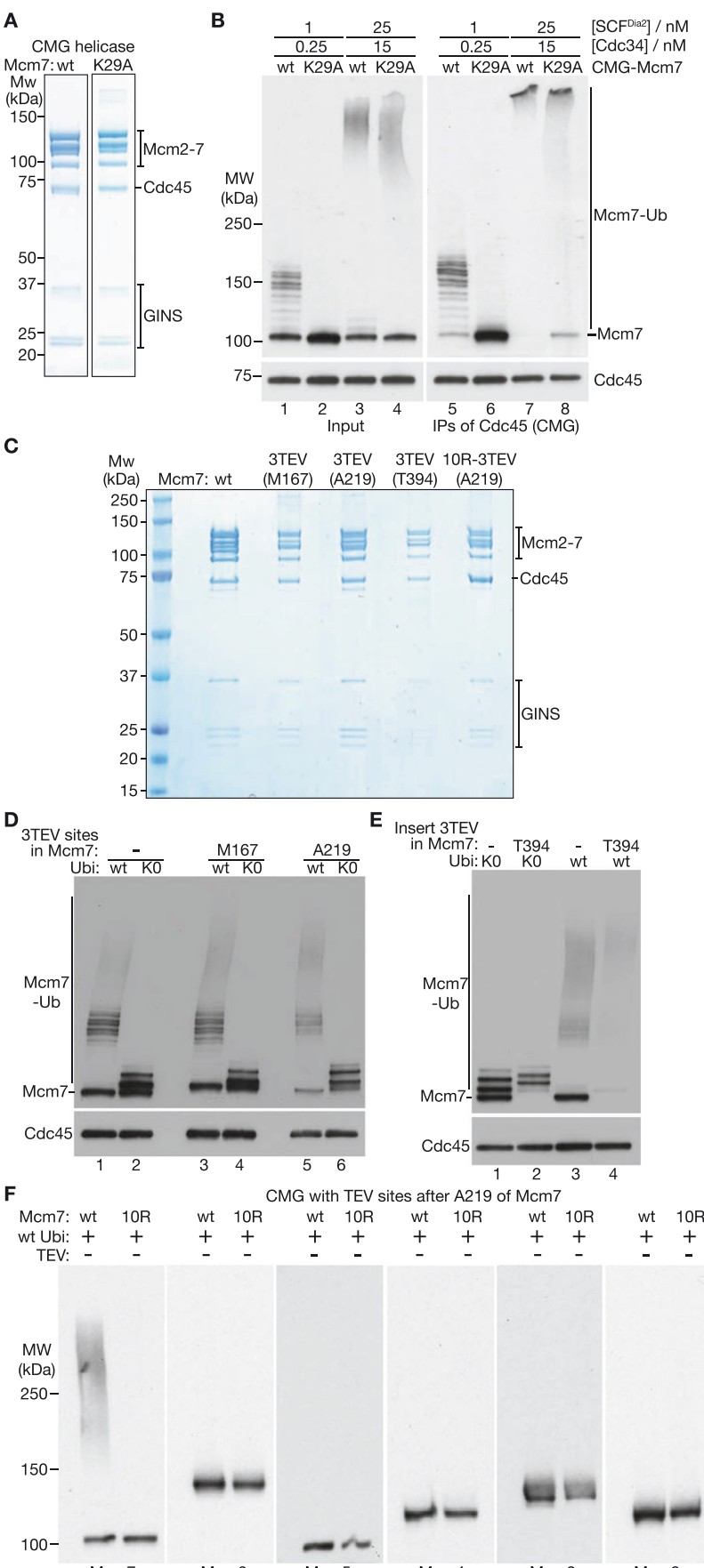

◀ **Figure EV3.  Generation and characterisation of recombinant CMG complexes with mutated alleles of Mcm7.**

(A) Purified recombinant budding yeast CMG helicase, with wild-type Mcm7 or Mcm7-K29A, were resolved by SDS-PAGE, before staining of the gel with Coomassie blue. (B) CMG containing wild-type (wt) Mcm7 or Mcm7-K29A was ubiquitylated in the presence of the indicated concentrations of SCF$^{Dia2}$ and Cdc34. The reactions were analysed by immunoblotting. (C) Coomassie-stained gel with purified recombinant CMG containing wild-type Mcm7 or the indicated variants. (D, E) Recombinant CMG with the indicated versions of Mcm7 were ubiquitylated in vitro by SCFDia2 and Cdc34, using either wild-type ubiquitin (wt Ubi) or lysine-free ubiquitin (K0 Ubi). Reactions were then analysed by immunoblotting. Note that the ubiquitylated forms in lanes 3–6 of (D), and lanes 2 and 4 of (E), are shifted by comparison with the control, due to the insertion into Mcm7 of peptide sequences containing TEV cleavage sites (the sites were not cleaved in these reactions). (F) Analogous reactions to those in D–E were performed with wild-type ubiquitin. The Mcm2–7 subunits of CMG were monitored by immunoblotting. Source data are available online for this figure.

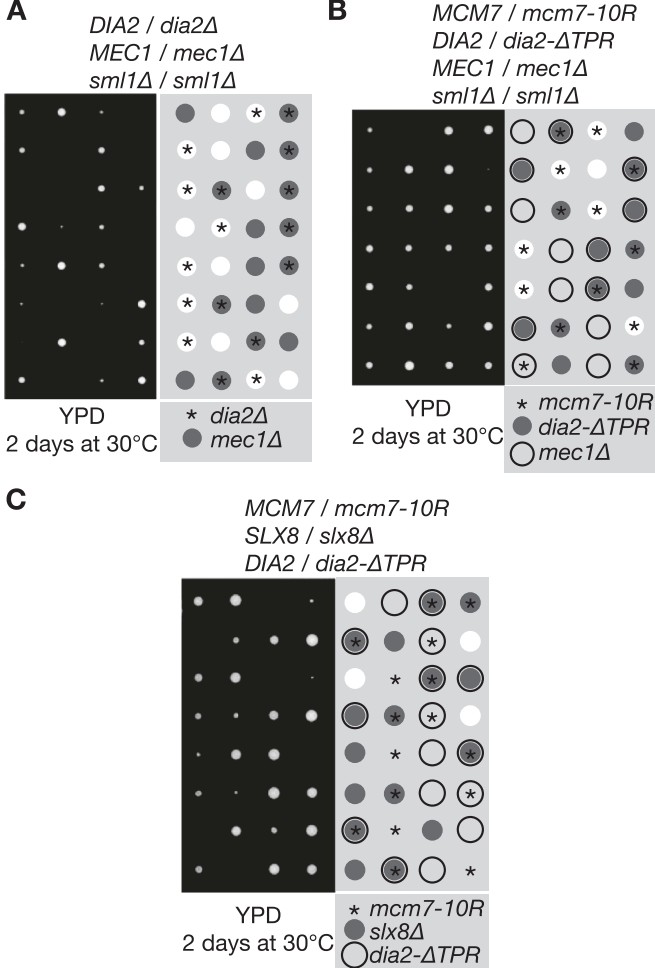

**A**

*DIA2 / dia2Δ*
*MEC1 / mec1Δ*
*sml1Δ / sml1Δ*

YPD
2 days at 30°C

★ *dia2Δ*
● *mec1Δ*

**B**

*MCM7 / mcm7-10R*
*DIA2 / dia2-ΔTPR*
*MEC1 / mec1Δ*
*sml1Δ / sml1Δ*

YPD
2 days at 30°C

★ *mcm7-10R*
● *dia2-ΔTPR*
○ *mec1Δ*

**C**

*MCM7 / mcm7-10R*
*SLX8 / slx8Δ*
*DIA2 / dia2-ΔTPR*

YPD
2 days at 30°C

★ *mcm7-10R*
● *slx8Δ*
○ *dia2-ΔTPR*

Figure EV4.  **The S-phase checkpoint and Slx8 become essential when the ubiquitylation of CMG-Mcm7 is blocked.**

(**A–C**) Tetrad analysis of diploid budding yeast cells of the indicated genotypes (YCPR486, YCPR500, YCPR222). YPD = medium comprising Yeast Extract, Peptone, Dextrose. Source data are available online for this figure.

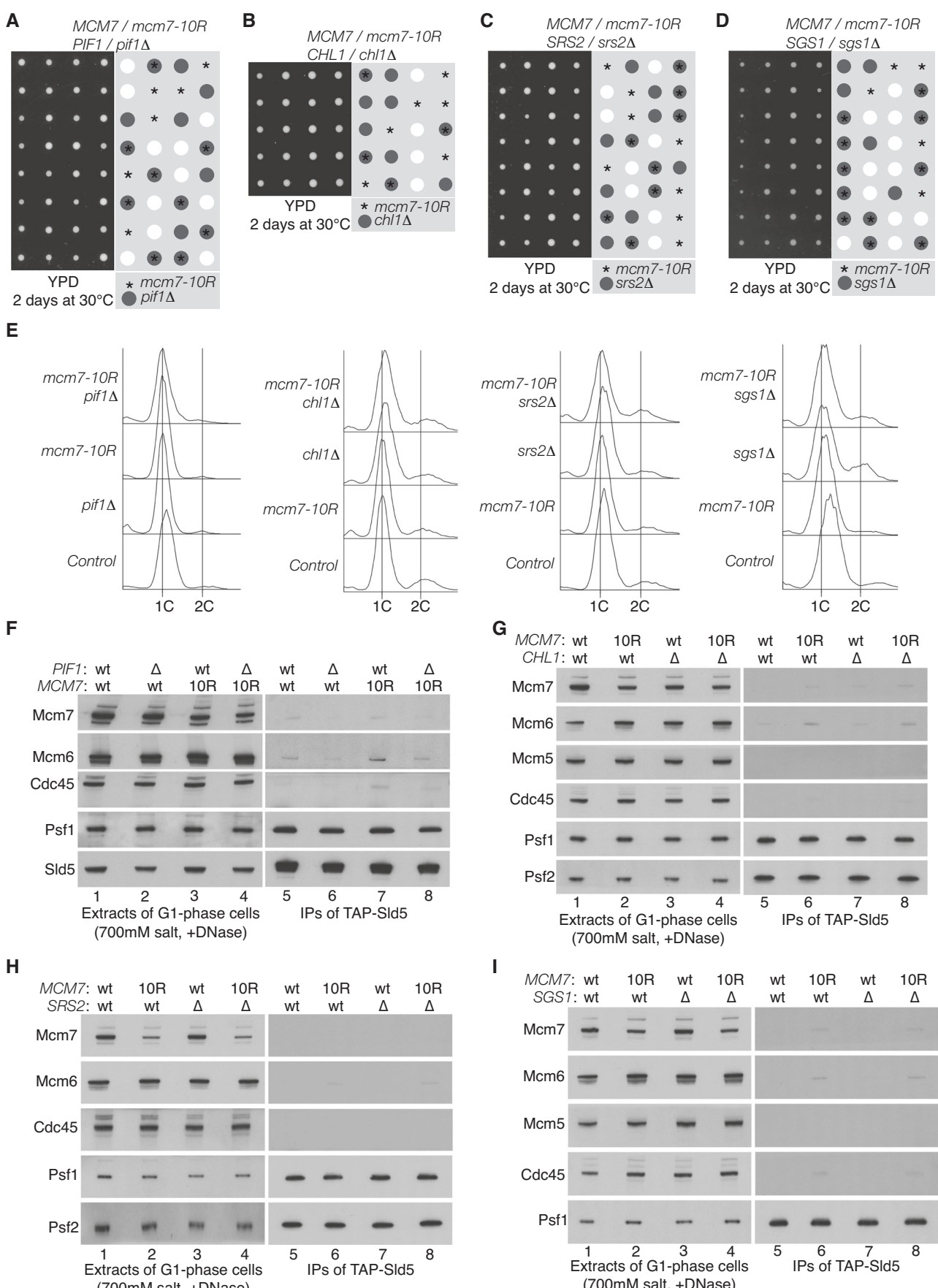

◀ **Figure EV5. Combination of *mcm7-10R* with *pif1*Δ, *chl1*Δ, *srs2*Δ or *sgs1*Δ does not cause synthetic lethality or accumulation of CMG during G1 phase.**

(A–D) Tetrad analysis of diploid budding yeast cells of the indicated genotypes (YCPR75, YCPR435, YCPR434, YCPR171). YPD = Yeast Extract, Peptone, Dextrose. (E) The indicated strains (YCPR141, YCPR406, YCPR412, YCPR428) were arrested in G1 phase at 30 °C by addition of mating pheromone. DNA content was monitored by flow cytometry. (F–I) Cell extracts were prepared from the samples in (E) and used to isolate GINS and the CMG helicase by immunoprecipitation of TAP-Sld5. The indicated factors were monitored by immunoblotting. Source data are available online for this figure.

