## [Peer Review File · The EMBO Journal]

CMG helicase disassembly is essential and mediated by two pathways in budding yeast

Karim Labib, Cristian Polo Rivera, and Tom Deegan

Corresponding author(s): Karim Labib (k.p.m.labib@dundee.ac.uk) , Tom Deegan (t.deegan@ed.ac.uk)

Review Timeline:

Submission Date:	24th Mar 24
Editorial Decision:	16th Apr 24
Revision Received:	19th May 24
Editorial Decision:	7th Jun 24
Revision Received:	12th Jun 24
Accepted:	19th Jun 24

Editors: Hartmut Vodermaier and Cornelius Schneider

Transaction Report:

Prof. Karim Labib
University of Dundee
MRC Protein Phosphorylation and Ubiquitylation Unit
Sir James Black Centre
Dow Street
Dundee DD1 5EH
United Kingdom

16th Apr 2024

Re: EMBOJ-2024-117387
CMG helicase disassembly is essential and driven by two pathways in budding yeast

Dear Karim,

Thank you again for submitting your study on the importance of CMG disassembly to The EMBO Journal. I have now received input from three expert referees, copied below for your information. All referees consider the studied question important and your findings potentially interesting, but they also raise various points that would need to be clarified prior to publication. Among those, referee 2's request to better differentiate old and new CMG complexes would appear very relevant for further strengthening the understanding of CMG unloading mechanisms.

Should you be able to satisfactorily respond to the issues raised by the referees, we would be happy to consider a revised version of the study further for The EMBO Journal. Given that we usually aim for only a single major-revision-round, I would encourage you to contact me with a tentative point-by-point response/revision plan already during the early stages of your revision work, on the basis of which I'd be happy to talk through the key points with you. We may also discuss an extension of the default three-months revision period if needed; our 'scooping protection' (meaning that competing work appearing elsewhere in the meantime will not affect our considerations of your study) would of course remain valid also throughout such an extension.

Further information on preparing, formatting and uploading a revised manuscript can be found below and in our Guide to Authors. Thank you again for the opportunity to consider this work for The EMBO Journal, and I look forward to hearing from you in due time.

With kind regards,

Hartmut

9) Digital image enhancement is acceptable practice, as long as it accurately represents the original data and conforms to community standards. If a figure has been subjected to significant electronic manipulation, this must be clearly noted in the figure legend and/or the 'Materials and Methods' section. The editors reserve the right to request original versions of figures and the original images that were used to assemble the figure. Finally, we generally encourage uploading of numerical as well as gel/blot image source data; for details see: embopress.org/page/journal/14602075/authorguide#sourcedata

At EMBO Press, we ask authors to provide source data for the main manuscript figures. Our source data coordinator will contact you to discuss which figure panels we would need source data for and will also provide you with helpful tips on how to upload and organize the files.

In the interest of ensuring the conceptual advance provided by the work, we recommend submitting a revision within 3 months (15th Jul 2024). Please discuss the revision progress ahead of this time with the editor if you require more time to complete the revisions. Use the link below to submit your revision:

Link Not Available

Referee #1:

This manuscript from the Labib lab examines the fate of replisome complexes that fail to be disassembled at the end of replication. The authors use clever biochemistry and genetics to generate the mcm7-10R mutant, which is refractory to the ubiquitin-dependent mechanism that normally unloads the replisome. mcm7-10R results in genome instability in the subsequent cell cycle and requires 5'-3' helicase activity for viability (either Rrm3 or Pif1). The requirement for a 5'-3' helicase is shared by dia2 mutants, which also cannot unload the replisome, suggesting this is a general requirement if ubiquitin-dependent replisome disassembly is blocked. The authors perform elegant experiments to show that Rrm3 promotes disassembly of replisome complexes that persist from the prior cell cycle, providing a potential explanation for its requirement in mcm7-10R/dia2 null cells.

Ever since the ubiquitin-dependent replisome disassembly pathway was discovered (Maric et al, Moreno et al) the question of why the replisome disassembly pathway is needed has sparked intense interest. Although it was quickly recognized that dia2 mutants are sick, specifically linking this back to replisome disassembly was challenging. Additionally, no obvious defect in replication was observed when replisome disassembly was blocked in biochemical settings. This is a truly awesome paper that identifies a crucial role for replisome disassembly in supporting faithful transmission of genetic material as well as cell viability. I have no major concerns about the manuscript and think it should be published swiftly.

Minor concerns

1. The Hill et al (2020) preprint should be mentioned in the introduction and more extensively in the discussion. I think the authors need to comment on whether the Pif1/Rrm3-dependent CMG disassembly mechanism they have described could be the

same mechanism that unloads pre-RCs (as in the Hill preprint).

2. The term 'metastable' is used but I think 'stable' or some other equivalent would be better because CMGs are retained for a long time if not actively disassembled.
3. Lines 103-106: I'm not sure I agree with this. Even if CMGs fail to disassemble, the total amount of CMGs per cell will halve during each mitosis. This alone will cause the CMGs to reach an equilibrium and not accumulate.
4. Lines 109-122: The second metazoan CMG disassembly pathway (TRAIP) should be mentioned here
5. EV2H: there should be a statement in the legend of exactly why the register of ubiquitinated species is shifted between lanes 1 and 2
6. EV2G-H: the legend should clearly state that these contain the 3TEV insert but are not TEV-treated
7. EV3B: there is no band for the +1 species so the legend should explain how the ubiquitinated species are assigned i.e. why lane 4 shows the +2 and +3 species and not +1 and +2.
8. Line 187: it's not immediately clear that the 624-845 fragment is less efficiently ubiquitinated than the 1-167 fragment because the modified species are of similar abundance. It would be helpful to direct the reader to the unmodified band in lanes 4 and 8 to better make this case.
9. Fig 2D: MCM5 appears reduced in lane 6. Is this just a loading issue?
10. Line 274: A|D should be A|D
11. Lines 290-294: to substantiate this it would be helpful to point out that in *dia2* mutants CMG does not undergo appreciable disassembly
12. Line 308: this appears synergistic, not additive
13. Lines 310-315: the *mcm7-10R* mutant alone appears to be slightly cold- and MMS-sensitive. Is this correct? If so, it should be mentioned. Currently the manuscript mentions only the phenotype of the *mcm7-10R dia2-TPR* double mutant.
14. Lines 303-336: is it possible that Tec1, Rad51, Sir4 and Cdc6 are not bona fide targets of SCFDia2 but instead are targeted due to their association with CMG? It would be appreciated if this could be commented on either way.
15. Lines 363-364: there needs to be a reference or justification for this statement
16. In Fig 5A it appears that *dia2 rrm3* and *dia2 pif1* mutants are basically inviable already. The accompanying text (371-374) focuses more on the lethality of the triple mutant, which is understandable since the synthetic sickness of *dia2* and *rrm3/pif1* has previously been published. Still, I think the dramatic synthetic sickness of the double mutants should be more clearly spelled out in the main text.
17. Line 385: why could the degron tag not be introduced? (Presumably due to lethality of the tag in this background?)
18. Lines 483-501: presumably either the C- or N- terminal face of an intact CMG could be encountered by a single fork. I don't understand why the authors envisage encounter with the C-terminal face to be different.
19. Line 518: I'm not sure twice counts as 'repeatedly'.

Non-essential suggestions

20. Lines 135-250: it's a little odd to start a manuscript with such extensive discussion of EV Figures. It would improve readability if this text were removed from the main manuscript. A cartoon (e.g. a modified form of EV2E) could be used to describe and summarize the results of EV2 and EV3 to assist with this.
21. The finding that 5'-3' helicases unload the replisome could have quite broad implications. Could a similar pathway operate in bacteria where there is currently no known replisome disassembly pathway? Could it also operate in human cells? The authors discuss the replisome recycling defects in LRR1-null human cells (Fan et al, 2021) but some of these defects could be due to CMGs left over from the previous cell cycle. Additionally, the TRAIP pathway still operates so replisome disassembly isn't totally blocked. I think it could be nice for the authors to discuss the potential for this pathway in humans and bacteria, but this is very much a non-essential suggestion.

Referee #2:

- General summary and opinion about the principal significance of the study, its questions and findings

The CMG helicase is the core component of eukaryotic replisome complex, and the mechanism of CMG assembly during replication initiation has been extensively and widely investigated in a lot of laboratories all over the world, whereas only a limited number of laboratories (including Prof. Labib's lab in this study) have paid attention to CMG disassembly process.

According to our knowledge, CMG disassembly requires poly-ubiquitylation of Mcm7 subunit by SCF E3 ligase. In this study, the authors confined ubiquitylation sites within 10 lysine residues in the N-terminal 219 amino acids region of Mcm7. Using the arginine mutant (*mcm7-10R*), they revealed the presence of additional CMG disassembly pathway. This second pathway is dependent on Pif1-family helicases (*Rrm3* and *Pif1*) but independent of the ubiquitylation system, and acts during S-phase to disassemble old CMG helicases from the previous cell cycle. The authors also demonstrated that yeast cells could not survive without either one of the two CMG disassembly pathways.

This study deals with an important issue in the DNA replication field, and the proposed model for disassembly of old and new CMGs is very interesting. I think the clear discrimination between old and new CMGs should be important for the conclusions without doubt, but only a mixture of new and old CMGs was analyzed in this study. It will be necessary to overcome this point for

publication in the EMBO journal.

- Specific major concerns essential to be addressed to support the conclusions

1. If the model in Figure EV5 is true, disassembly of old CMG needs collision with new CMG or replication fork. In most cases of this study, G1-arrested cells were released to progress into S-phase and further forward. Although DNA replication progression was monitored by FACS, it was unknown whether old CMG disassembly is dependent on replication initiation or new CMG formation. Is it possible to show such a dependency?
2. To establish the model, it will be important to clearly distinguish old CMG from new CMG, for example, by expressing two types of differentially tagged Sld5s in order (although this may be difficult to perform within a short period). Alternatively, it will be meaningful to clarify different features of old CMG, if any. It may be possible that other replisome factors such as Mrc1/Ctf4/Pole no longer associate with old CMG after M-phase progression. The presence or absence of those factors may affect the residual ubiquitylation-dependent pathway in the *mcm7-10R* cells during prolonged G1-phase.
3. In relation with this, as the authors described at the line 391-394, only a proportion of old CMG remained in G1 in the *mcm10-10R rrm3Δ* cells (Fig 5F), and a large proportion was removed either due to residual ubiquitylation ---, or else reflecting the ability of additional factors ---. One problem is that it is unclear when and how these residual activities may act for old and new CMGs. The asynchronous *mcm10-10R rrm3Δ* culture should have consisted of G1-cells with old CMG, S-cells with ongoing (new) and old CMGs, and G2/M cells with old CMG as indicated by the abundance of CMG (Fig. 5F). At this point, the nature of old CMG was unclear. Since the two pathways were not fully functional, new CMG would become old CMG while pre-existed old CMG would also remain in the following G2. Therefore, this culture might contain more than two rounds of old CMGs even when arrested in G1. This may be the same situation with Figure 6 (2nd G1). Nevertheless, a large proportion of those old CMG was removed by the above mentioned residual activities. If the authors could address only the fate of old CMG (that may be labeled with a specific tag or so) in the condition of Figure 7, it will produce very convincing results. This analysis might also be performed with the *dia2Δ rrm3Δ* (or *mcm7-10R TPRΔ rrm3Δ*) strain if available, because *dia2Δ* (or *mcm7-10R TPRΔ*) would leave much more old CMG even in prolonged G1 (Fig. 4B).

- Minor concerns that should be addressed

1. At the line 262-263, it was stated that "*cdc48-aid* and *cdc48-AID mcm7-10R* cells were released from --- ". Is there any difference between the genotypes of *aid* and *AID*?
2. In the in vitro disassembly assay of Figure 2D, why did not Cdc45 dissociate from GINS? Did the other replisome factors (Mrc1/Ctf4/Pole) dissociate in the same condition?
3. In Figure 6B, there may be a mistake in the time courses, 10' release/20'/30'/40'/50'/40'/80'/120' ---. The 2nd 40' will be correctly 60'.
4. Is the Rrm3/Pif1-mediated pathway for old CMG dependent on Top2 activity?
5. Does the Rad52 foci in *mcm7-10R* increase or decrease in the absence of Rrm3?

- Any additional non-essential suggestions for improving the study (which will be at the author's/editor's discretion)

1. If possible, it might be better to shorten the description of identifying the ubiquitylation domain of Mcm7 (Fig. EV2&3) in the first section of 'Results' part.
2. At the line 235, it might be better to clearly state the names of 'other replisome proteins' (Ctf4, Mrc1, Pole) not only in the figure legend but also in the text.
3. At the line 363-364, it was stated that "When *dia2Δ* cells progress through S-phase, old CMG complexes from the previous cell cycle are removed by an unknown pathway." Since this is just a speculation at that point, it is better to add "(Fig EV1B).
4. Is it possible to remove old CMG in *dia2Δ* cells by exogenously expressed Dia2 in G1-phase?

Referee #3:

In this manuscript, Rivera and colleagues report the characterization of a fundamental mechanism by which eukaryotes unload the replicative helicase CMG from chromatin upon termination of DNA replication. Previous studies from the Labib lab and from others have shown that converging CMG helicases bypass each other at replication termination sites and remain topologically trapped around DNA, until they are ubiquitylated and displaced from chromatin by Cdc48. In budding yeast, the ubiquitylation of the CMG subunit MCM7 depends on the ubiquitin ligase SFCDia2. Failure to disassemble CMG complexes in the absence of Dia2 results in genomic instability and growth defects. Because SFCDia2 targets several other DNA replication and repair factors, the deleterious effect of the persistence of trapped CMG complexes on DNA has not been formally established. Here, the authors used recombinant CMG and a reconstituted ubiquitylation assay to identify critical Mcm7 residues targeted by Dia2. With this novel information in hand, they derived a largely non-ubiquitylatable version of Mcm7 (Mcm7-10R), which was used for further in vitro and in vivo studies. They show that cells expressing this allele phenocopy *dia2* mutants, especially when combined with the ablation of the Dia2 TPR domain, involved in its recruitment to the replisome. Interestingly, *Mcm7-10R* cells accumulated CMG on DNA after DNA replication but were viable due to a second pathway involving helicases of the Pif1 family. In particular, the authors show that the Rrm3 helicase is essential for viability in the absence of Mcm7 ubiquitylation, and

that Rrm3 exerts this critical function during DNA replication by removing 'old' CMG complexes that remain on chromatin after the previous S phase.

Overall, the data presented in the manuscript are of high quality and support the view that two distinct pathways coexist in eukaryotic cells to degrade CMG helicases after completion of DNA replication. The first pathway is an ancestral pathway that relies on a conserved family of DNA helicases to remove old CMG complexes during DNA synthesis. The second pathway depends on less conserved enzymes that ubiquitylate CMG at the end of the S phase to promote its degradation by Cdc48/p97 and the proteasome. These findings represent a major advance in our understanding of this important cellular process. However, specific issues need to be addressed to strengthen their case.

1. The first third of the manuscript describes the strategy used to identify the major ubiquitylation sites of Mcm7, which were then mutated to generate the Mcm7-10R construct. The strategy used to identify these sites is very elegant and it is a pity that at least some of these data are not presented in a main figure.
2. Two figures are listed as Appendix Figures (S1 and S2), but Appendix Figures S1 is not even mentioned. These figures should be listed as EV figures so as not to unnecessarily complicate the structure of the manuscript.
3. Line 257: should read (Fig 3A and 3D).
4. Line 263: The authors should explain why Cdc48 is depleted in HU-arrested cells and not in late G1 (Fig 3A). Would this interfere with the G1/S transition?
5. In the experiment shown in Figure 3D, cells were synchronously released from an alpha-factor arrest into S phase. However, a large fraction of the cell population (~40%) appears to remain blocked in G1. This is surprising because these cells correctly degrade the CDK inhibitor Sic1. Although this does not affect the message of the figure, the authors should provide an experiment showing complete release from G1 arrest, as is the case in Figure 3A.
6. Line 370: The authors should also cite a recent article from the Aharoni lab showing that Rrm3 unwinds G4 on the leading strand while Pif1 acts on the lagging strand (PMID: 38117984).
7. In Fig EV5, the authors present different scenarios of CMG disassembly in different contexts, which are further discussed at the end of the Discussion section. It has recently been proposed by the Cortez lab that CMG is trapped on DNA by RAD51 during fork reversal, while the rest of the replisome disassembles. In such a configuration, would a converging fork disassemble this CMG in a manner compatible with proper termination of DNA replication?

“Referee #1:

This manuscript from the Labib lab examines the fate of replisome complexes that fail to be disassembled at the end of replication. The authors use clever biochemistry and genetics to generate the mcm7-10R mutant, which is refractory to the ubiquitin-dependent mechanism that normally unloads the replisome. mcm7-10R results in genome instability in the subsequent cell cycle and requires 5'-3' helicase activity for viability (either Rrm3 or Pif1). The requirement for a 5'-3' helicase is shared by dia2 mutants, which also cannot unload the replisome, suggesting this is a general requirement if ubiquitin-dependent replisome disassembly is blocked. The authors perform elegant experiments to show that Rrm3 promotes disassembly of replisome complexes that persist from the prior cell cycle, providing a potential explanation for its requirement in mcm7-10R/dia2 null cells.

Ever since the ubiquitin-dependent replisome disassembly pathway was discovered (Maric et al, Moreno et al) the question of why the replisome disassembly pathway is needed has sparked intense interest. Although it was quickly recognized that dia2 mutants are sick, specifically linking this back to replisome disassembly was challenging. Additionally, no obvious defect in replication was observed when replisome disassembly was blocked in biochemical settings. This is a truly awesome paper that identifies a crucial role for replisome disassembly in supporting faithful transmission of genetic material as well as cell viability. I have no major concerns about the manuscript and think it should be published swiftly.”

We thank the referee for his / her interest in our work and for the helpful comments below.

Minor concerns

“1. The Hill et al (2020) preprint should be mentioned in the introduction and more extensively in the discussion. I think the authors need to comment on whether the Pif1/Rrm3-dependent CMG disassembly mechanism they have described could be the same mechanism that unloads pre-RCs (as in the Hill preprint).”

We now include further citations to Hill et al in the Introduction (lines 61-62) and in the Discussion (lines 500-502).

In addition, we discuss the relation between the disassembly pathways for old CMG and pre-RCs on lines 529-534, again citing the Hill manuscript.

“2. The term 'metastable' is used but I think 'stable' or some other equivalent would be better because CMGs are retained for a long time if not actively disassembled.”

We have now removed metastable and instead use 'stable' (line 14) or 'highly stable' (line 90).

“3. Lines 103-106: I'm not sure I agree with this. Even if CMGs fail to disassemble, the total amount of CMGs per cell will halve during each mitosis. This alone will cause the CMGs to reach an equilibrium and not accumulate.”

We take the referee's point that mitosis will slow the accumulation of old CMG in *dia2Δ* cells, yet old CMG complexes should still accumulate gradually in the absence of a second pathway for disassembly. If a cell ends S-phase with 'X' amount of CMG (in CMG disassembly were inactivated in a single cell cycle), the daughter cells would enter the next cycle with 0.5X and end S-phase with 1.5X. The following mitosis would reduce the level to 0.75X, which would then increase to 1.75X in the next S-phase, etc.

"4. Lines 109-122: The second metazoan CMG disassembly pathway (TRAIP) should be mentioned here"

We now cite the TRAIP pathway on lines 123-124 and line 130.

"5. EV2H: there should be a statement in the legend of exactly why the register of ubiquitinated species is shifted between lanes 1 and 2"

Thank you for pointing this out - we now explain in the corresponding figure legend (lines 1274-1277) that this is due to the insertion into Mcm7 (lanes 2 and 4) of a peptide sequence containing TEV cleavage sites. These data are now located in Fig EV3E.

"6. EV2G-H: the legend should clearly state that these contain the 3TEV insert but are not TEV-treated"

Once again, thank you for pointing this out and we now explain this in the figure legend (line 1277) for what is now Fig EV3D-E.

"7. EV3B: there is no band for the +1 species so the legend should explain how the ubiquitinated species are assigned i.e. why lane 4 shows the +2 and +3 species and not +1 and +2."

In the revised version, we now deal with this issue in the legend to what is now Fig 1A (lines 1070-1072), which states:

"Where necessary, the identity of the various ubiquitylated forms was determined via longer exposures of the same immunoblot (e.g. for the +1 form in lane 4)."

"8. Line 187: it's not immediately clear that the 624-845 fragment is less efficiently ubiquitinated than the 1-167 fragment because the modified species are of similar abundance. It would be helpful to direct the reader to the unmodified band in lanes 4 and 8 to better make this case."

We now label the unmodified band ("0") in all panels for this experiment (the data are now in Fig 1A-E) to make this clearer.

"9. Fig 2D: MCM5 appears reduced in lane 6. Is this just a loading issue?"

Probably – unfortunately we don't have any sample remaining to rerun the gel.

"10. Line 274: A|D should be AID"

If the referee meant that we sometimes used lowercase and sometimes upper case, we have now corrected this and used uppercase throughout, to be more consistent.

"11. Lines 290-294: to substantiate this it would be helpful to point out that in *dia2* mutants CMG does not undergo appreciable disassembly"

We now make this point on lines 307-309.

“12. Line 308: this appears synergistic, not additive”

We now make this change on line 325.

“13. Lines 310-315: the *mcm7-10R* mutant alone appears to be slightly cold- and MMS-sensitive. Is this correct? If so, it should be mentioned. Currently the manuscript mentions only the phenotype of the *mcm7-10R dia2-TPR* double mutant.”

We now explain on lines 332-333 that the *mcm7-10R* single mutant causes mild sensitivity to cold (Fig 5C) and MMS (Fig 5D).

“14. Lines 303-336: is it possible that *Tec1*, *Rad51*, *Sir4* and *Cdc6* are not bona fide targets of *SCFDia2* but instead are targeted due to their association with CMG? It would be appreciated if this could be commented on either way.”

This seems unlikely, as these factors have not been found to associate with CMG. We comment on this point on lines 339-340.

“15. Lines 363-364: there needs to be a reference or justification for this statement”

We have rephrased this section (lines 382-384), to refer to the earlier discussion of this issue in the Introduction (lines 117-120).

“16. In Fig 5A it appears that *dia2 rrm3* and *dia2 pif1* mutants are basically inviable already. The accompanying text (371-374) focuses more on the lethality of the triple mutant, which is understandable since the synthetic sickness of *dia2* and *rrm3/pif1* has previously been published. Still, I think the dramatic synthetic sickness of the double mutants should be more clearly spelled out in the main text.”

We have re-written the section on lines 385-386 to make this point clear.

“17. Line 385: why could the degron tag not be introduced? (Presumably due to lethality of the tag in this background?)”

We were able to make degron-tagged strains for *pif1* and *rrm3* – the problem was that the small amount of degron-tagged protein that remained upon depletion was still sufficient to supply function (this is a very common issue with protein depletion systems in all species).

We now refer to this more directly on lines 401-402:

“*this was not successful due to incomplete degradation of degron-tagged protein*”.

“18. Lines 483-501: presumably either the C- or N- terminal face of an intact CMG could be encountered by a single fork. I don't understand why the authors envisage encounter with the C-terminal face to be different.”

When two forks converge during termination in wild type cells, Rrm3-Pif1 are unable to cause CMG disassembly, despite the potential for Rrm3-Pif1 associated with the lagging strand template at one fork to encounter CMG on the leading strand template from the opposing fork. Under such conditions, Rrm3-Pif1 would always encounter the N-terminal Mcm2-7 face of CMG (e.g. see Fig 9B (iii) of the revised version). It is possible, therefore, that Rrm3-Pif1 cannot disassemble CMG when they encounter the N-terminal Mcm2-7 face of CMG and instead need to meet the C-terminal Mcm2-

7 face to drive CMG disassembly. This would be compatible with Rrm3-Pif1 disassembling old CMG in the second S-phase, since one of the two forks that converge on an old CMG would always meet the C-terminal Mcm2-7 face (the other would of course meet the N-terminal Mcm2-7 face). However, for simplicity we have now removed this suggestion and rewritten the discussion of possible mechanisms (lines 519-534).

“19. Line 518: I'm not sure twice counts as 'repeatedly'.”

We agree! However, the idea was that CMG ubiquitylation would have evolved at least three times, should plants also ubiquitylate CMG during DNA replication termination. At present, we have evidence for CMG ubiquitylation having evolved twice in fungi and animals, involving ligases that are absent in plants. Since the situation in plants is currently unknown (they either use a different ligase, or else disassemble CMG via a different mechanism), we have now rephrased the text on lines 551-552 to say that the existence of different ligases in fungi and animals “suggests that CMG ubiquitylation might have arisen multiple times during eukaryotic evolution”.

Non-essential suggestions

“20. Lines 135-250: it's a little odd to start a manuscript with such extensive discussion of EV Figures. It would improve readability if this text were removed from the main manuscript. A cartoon (e.g. a modified form of EV2E) could be used to describe and summarize the results of EV2 and EV3 to assist with this.”

We thank the referee for these helpful comments and agree that it would be better not to start with an extensive discussion of EV figures. We also liked the idea of including a cartoon (based on what was EV2E) to summarise the results. At the same time, Referee 3 suggested moving these data to a main figure. Therefore, we have combined these suggestions in the revised manuscript. The ‘TEV-mapping’ experiments are now presented in a new Figure 1, which allows us to include visual aids such as the alignment and the structure illustrating the location in Mcm7 of the TEV cleavage. We also include a new cartoon in Fig 1F, which summarises the results of the TEV-mapping experiments.

21. The finding that 5'-3' helicases unload the replisome could have quite broad implications. Could a similar pathway operate in bacteria where there is currently no known replisome disassembly pathway? Could it also operate in human cells? The authors discuss the replisome recycling defects in LRR1-null human cells (Fan et al, 2021) but some of these defects could be due to CMGs left over from the previous cell cycle. Additionally, the TRAIP pathway still operates so replisome disassembly isn't totally blocked. I think it could be nice for the authors to discuss the potential for this pathway in humans and bacteria, but this is very much a non-essential suggestion.

We agree that bacterial replisome disassembly remains an interesting unsolved question. Bacterial orthologues of Pif1 do exist but might not be expected to disassemble the replicative helicase, which in bacteria has a 5' to 3' polarity (the same as Pif1) and thus occupies the same template strand as Pif1. The question of whether Pif1 can act to remove CMG in humans is also interesting. At this stage we can say very little, given the presence of the TRAIP pathway that

mediates CMG disassembly during mitosis, and the presence of other 5' to 3' helicases such as RTEL1 that might have taken over some of the roles of yeast Rrm3 / Pif1.

These would probably be interesting issues to discuss in a review article, but for now we think that we don't really have space to discuss them further in our manuscript, beyond the statement in the final sentence of the Discussion that "***It also remains to be determined whether Pif1-family helicases mediate CMG helicase disassembly in other species apart from budding yeast, and whether this role is shared with other 5' to 3' DNA helicases that help replication forks to bypass roadblocks, such as the RTEL1 helicase in metazoa (Hourvitz et al, 2023; Vannier et al, 2014).***"

In the context of the data in our manuscript, we felt that the major implication to discuss at this stage was the fact that Pif1 helicases are great candidates for an ancient pathway of CMG disassembly in ancestral eukaryotes, before the evolution of CMG disassembly in fungi, animal cells (and maybe plants too).

Referee #2:

“- General summary and opinion about the principal significance of the study, its questions and findings

The CMG helicase is the core component of eukaryotic replisome complex, and the mechanism of CMG assembly during replication initiation has been extensively and widely investigated in a lot of laboratories all over the world, whereas only a limited number of laboratories (including Prof. Labib's lab in this study) have paid attention to CMG disassembly process.

According to our knowledge, CMG disassembly requires poly-ubiquitylation of Mcm7 subunit by SCF E3 ligase. In this study, the authors confined ubiquitylation sites within 10 lysine residues in the N-terminal 219 amino acids region of Mcm7. Using the arginine mutant (mcm7-10R), they revealed the presence of additional CMG disassembly pathway. This second pathway is dependent on Pif1-family helicases (Rrm3 and Pif1) but independent of the ubiquitylation system, and acts during S-phase to disassemble old CMG helicases from the previous cell cycle. The authors also demonstrated that yeast cells could not survive without either one of the two CMG disassembly pathways.

This study deals with an important issue in the DNA replication field, and the proposed model for disassembly of old and new CMGs is very interesting. I think the clear discrimination between old and new CMGs should be important for the conclusions without doubt, but only a mixture of new and old CMGs was analyzed in this study. It will be necessary to overcome this point for publication in the EMBO journal.”

We thank the Referee for her / his comments and are glad to see that he / she thought that our work “*deals with an important issue in the DNA replication field, and the proposed model for disassembly of old and new CMGs is very interesting*”.

We agree entirely with the Referee that the clear discrimination between old and new CMGs is important. For this reason, we have spent a lot of time exploring this issue, both in our previously published work (Maric et al, 2014, *Science*) and via additional new data, as discussed in detail below.

“- Specific major concerns essential to be addressed to support the conclusions

1. If the model in Figure EV5 is true, disassembly of old CMG needs collision with new CMG or replication fork. In most cases of this study, G1-arrested cells were released to progress into S-phase and further forward. Although DNA replication progression was monitored by FACS, it was unknown whether old CMG disassembly is dependent on replication initiation or new CMG formation. Is it possible to show such a dependency?”

As discussed below, a combination of previous work, the data in the first version of our submitted manuscript, new data in the revised version and other new work cited therein, strongly support the idea that Rrm3-Pif1 cannot disassemble old CMG from DNA replication termination until entry into S-phase of the following cell cycle, at which point Rrm3-Pif1 function at replication forks in a manner dependent upon interactions with the replisome. Further progress beyond this point will be dependent upon the future reconstitution in vitro of the clash between new replication forks and old CMG, which goes beyond the scope of our current study.

Work from other groups (e.g. Lahaye et al, J. Biol. Chem., 268, 26155-61, 1993) showed that Pif1-family helicases are greatly stimulated by forked DNA substrates. This provides a mechanistic explanation for why Rrm3 / Pif1 cannot displace the old CMG complexes that encircle double-strand DNA after DNA replication termination, until the arrival of a new DNA replication fork at the old CMG complex, in the next round of S-phase. Correspondingly, the ability of Pif1 to displace MCM2-7 double hexamers from plasmid DNA in vitro is dependent upon the presence of replication forks (Hill et al, bioRxiv, 2020). These data are now cited in the revised version of our manuscript (e.g. lines 499-502).

Consistent with the above, our previously published data (Maric et al, 2014, *Science*) indicates that post-termination CMG cannot be disassembled by Rrm3 / Pif1 within the same cell cycle, or even into G1-phase of the next cell cycle. Using a similar approach to the one suggested below by Referee 2, we compared control cells with *dia2Δ* cells (in which the ubiquitylation pathway for CMG disassembly is completely blocked). Cells were arrested in G1-phase, before expression of either a tagged GINS subunit (Figure 5B, Maric et al, 2014), or a tagged form of Cdc45 (Figure S7-S8, Maric et al, 2014). Cells were then released from G1-arrest and allowed to progress through S-phase / G2-phase / mitosis / G1-phase of the following cell cycle. In this way, we observed that CMG assembly in *dia2Δ* is comparable to wild type cells. However, we found that old CMG (CMG with tagged GINS or Cdc45, present in *dia2Δ* cells at timepoints after CMG had already disappeared in control cells) cannot be disassembled throughout S-phase / G2-phase / M-phase / G1-phase, despite the presence of Rrm3 and Pif1 (Figure S8 of Maric et al, 2014, shows the longest timecourse that extends into G1-phase of the next cell cycle). It is important to note that old CMG cannot be monitored during S-phase of the second cell cycle in such experiments, in which tagged CMG components are expressed during G1-phase of the first cell cycle, since the tagged forms of GINS / Cdc45 would then be present in a mixture of old and new CMG complexes.

These findings strongly indicate that Rrm3 / Pif1 cannot disassemble old CMG complexes between S-phase of one cell cycle and the end of the following G1-phase. Correspondingly, our data in the present manuscript (Figure 7C, lanes 5-6) show that Rrm3 cannot disassemble old CMG helicase complexes during G1-phase of the following cell cycle (in this experiment, the observed CMG complexes are unambiguously 'old', as CMG assembly cannot occur during G1-phase). In contrast, old CMG complexes from the previous cell cycle are disassembled when cells enter S-phase and pass through the next cell cycle (Figure 7C, compare lanes 6 and 8), dependent upon the presence of Rrm3 (Figure 7C, compare lanes 7 and 8). The simplest explanation of these data, considered together with the previous

biochemical studies of Pif1 helicases referred to above, is that Rrm3 functions at DNA replication forks during the second S-phase, to disassemble old CMG complexes from the first S-phase.

Consistent with this idea, a new study from Tom Deegan's lab (Olson et al, under revision at *The EMBO Journal*) demonstrates that Rrm3 function is dependent upon its interaction at replication forks with the CMG helicase and DNA polymerase ϵ . Mutations in Rrm3 that break these interactions severely impair the growth of *dia2 Δ* cells, resembling the previously observed synthetic lethality of *rrm3 Δ* with *dia2 Δ* , and likely reflecting a near-lethal defect in CMG helicase disassembly. These data are now cited in our revised manuscript (lines 510-511) and support the idea that Rrm3 functions at DNA replication forks to disassemble old CMG. The Olson manuscript is included with our revised submission, to allow the Referee to see the relevant data.

Finally, we note that the data originally in Figure 4G of our submitted manuscript (now in Figure 5G of the revised version) indicated that a defect in disassembling old CMG complexes during one cell cycle leads to genome instability (monitored via Rad52-GFP foci) in the next cell cycle. We subjected *mcm7-10R* cells to an extended G1 arrest (to remove most of the old CMG from previous cycles and allow cells to become enlarged) and then monitored them throughout the next two cell cycles. Rad52-GFP foci specifically increased in *mcm7-10R* cells that had completed the first cell cycle and rapidly entered S-phase after a very short G1-phase in the second cell cycle (i.e. binucleate cells that had undergone cytokinesis and then budded before completing cell separation – this is now illustrated in Fig 5G and described in the figure legend). This demonstrates that *mcm7-10R* cells experience DNA damage (reflected by Rad52-GFP foci) when they enter S-phase of the second cell cycle in the presence of old CMG. These data are consistent with the model that old CMG complexes are disassembled by the Rrm3-Pif1 pathway during the second S-phase. Such CMG disassembly comes at a price (some degree of DNA damage), which likely contributed to the evolution of ubiquitylation pathways that can disassemble old CMG complexes in the first cell cycle, immediately after DNA replication termination.

The referee might have envisioned an experiment in which we would block initiation (e.g. via degron-tagged Sld3) and then test whether this blocks the disappearance of old CMG. We have given much thought to such experiments in the past, but we don't think that it's possible to design a good control (in which we don't block initiation). For example, if the control represents *mcm7-10R* cells that contain old CMG during G1-phase, then the old CMG complexes should be disassembled when cells enter S-phase and new forks meet old CMG. However, this is very hard to monitor specifically, as total CMG would increase (due to new CMG assembly), and if old CMG carried a tag on GINS or Cdc45, then the same tagged version of the protein would also be incorporated into new CMG (since we wouldn't be blocking initiation in the control). Therefore, the 'experiment' might work (we might see that tagged CMG did not disappear when we blocked initiation), but the control would not work (tagged CMG would also persist in control cells, due to new CMG formation during initiation).

In summary, we hope the referee might agree that the data discussed above mean that by far the simplest interpretation of the present and past data, is that Rrm3-Pif1 act during S-phase to disassemble old CMG via forks. Proving this directly is a very interesting future challenge that in our view will require in vitro reconstitution of such reactions. We look forward to seeing someone do this!

“2. To establish the model, it will be important to clearly distinguish old CMG from new CMG, for example, by expressing two types of differentially tagged Sld5s in order (although this may be difficult to perform within a short period). Alternatively, it will be meaningful to clarify different features of old CMG, if any. It may be possible that other replisome factors such as Mrc1/Ctf4/Polε no longer associate with old CMG after M-phase progression. The presence or absence of those factors may affect the residual ubiquitylation-dependent pathway in the mcm7-10R cells during prolonged G1-phase.”

Please see above for discussion of our previously published data that distinguish old and new CMG in *dia2Δ* cells (Maric et al, 2014, Science), indicating that Rrm3 / Pif1 cannot disassemble old CMG between S-phase of the first cell cycle and the end of G1-phase in the second cell cycle.

Regarding the partners of old CMG in G1-phase cells, we previously showed in Figure 4B of Maric et al 2014 that old CMG complexes in *dia2Δ* cells can still associate during G1-phase with other replisome components such as Ctf4 and Tof1-Csm3. Our further unpublished analysis of the material from the same experiment shows that old CMG in *dia2Δ* cells also associates during G1-phase with the Pol2 catalytic subunit of DNA polymerase epsilon (see ‘Figure 1 for Referees Only’). This indicates that old CMG complexes can still interact during G1-phase with other replisome factors that are required for efficient CMG disassembly (Deegan et al, 2020).

Correspondingly, we showed in Maric et al 2014 that re-expression of Dia2 during G1-phase in *dia2Δ* cells induces rapid and efficient disassembly of old CMG complexes (Figure 6B of Maric et al 2014; compare samples 2 and 3). We now refer to these data on lines 110-113, and the data provide functional proof that old CMG complexes are fully competent, even during G1 phase, for ubiquitylation by SCF-Dia2 and disassembly by Cdc48.

“3. In relation with this, as the authors described at the line 391-394, only a proportion of old CMG remained in G1 in the mcm10-10R rrm3Δ cells (Fig 5F), and a large proportion was removed either due to residual ubiquitylation ---, or else reflecting the ability of additional factors ---. One problem is that it is unclear when and how these residual activities may act for old and new CMGs. The asynchronous mcm10-10R rrm3Δ culture should have consisted of G1-cells with old CMG, S-cells with ongoing (new) and old CMGs, and G2/M cells with old CMG as indicated by the abundance of CMG (Fig. 5F). At this point, the nature of old CMG was unclear. Since the two pathways were not fully functional, new CMG would become old CMG while pre-existed old CMG would also remain in the following G2. Therefore, this culture might contain more than two rounds of old CMGs even when arrested in G1. This may be the same

situation with Figure 6 (2nd G1). Nevertheless, a large proportion of those old CMG was removed by the above mentioned residual activities. If the authors could address only the fate of old CMG (that may be labeled with a specific tag or so) in the condition of Figure 7, it will produce very convincing results. This analysis might also be performed with the *dia2Δ rrm3Δ* (or *mcm7-10R TPRΔ rrm3Δ*) strain if available, because *dia2Δ* (or *mcm7-10R TPRΔ*) would leave much more old CMG even in prolonged G1 (Fig. 4B)."

As noted above:

- SCF-Dia2 is fully able to drive CMG ubiquitylation at any stage of the cell cycle, including G1-phase (as we showed previously in Maric et al, 2014)
- Rrm3 / Pif1 are unable to disassemble old CMG in *dia2Δ* cells, between S-phase when such old CMG complexes are formed, and the end of the following G1-phase, and yet are then able to drive CMG disassembly upon entry into the subsequent S-phase.
- we previously used tagged versions of Cdc45 and GINS proteins (Maric et al, 2014, *Science*) to show that old CMG in *dia2Δ* cells is not disassembled between DNA replication termination and the end of G1-phase in the next cell cycle, despite the presence of Rrm3-Pif1.

We have also performed similar experiments with tagged Cdc45 in *mcm7-10R rrm3Δ* cells, as suggested by the referee ('Figure 2 for Referees only'). However, this only allowed us to monitor CMG in a single cell cycle (we can follow the assembly of the new CMG and then its persistence after S-phase as old CMG). We cannot monitor old CMG during S-phase of the second cell cycle in such experiments, since the tagged forms of Cdc45 would then be present in a mixture of old and new CMG complexes. Also, it wouldn't work to have GAL-tagged CMG component and GAL-RRM3 in the same experiment (the CMG component would need to be induced a cell cycle earlier than Rrm3, which wouldn't be possible).

The referee also suggested looking at *dia2Δ rrm3Δ* (or *mcm7-10R dia2-ΔTPR rrm3Δ*), but these combinations are synthetic lethal and so cannot be analysed in an analogous fashion to the *mcm7-10R rrm3Δ* strain (the latter is viable due to the hypomorphic nature of the CMG disassembly defect in *mcm7-10R*).

- Minor concerns that should be addressed

"1. At the line 262-263, it was stated that "*cdc48-aid* and *cdc48-AID mcm7-10R* cells were released from --- ". Is there any difference between the genotypes of *aid* and *AID*?"

Thanks for pointing this out – it was just a typo and we now use 'AID' throughout.

"2. In the *in vitro* disassembly assay of Figure 2D, why did not Cdc45 dissociate from GINS? Did the other replisome factors (*Mrc1/Ctf4/Polε*) dissociate in the same condition?"

This reflects residual association of GINS and Cdc45 after CMG disassembly, as seen previously (for example, Figure 5A of Deegan et al, 2020, eLife).

Other replisome factors cannot be monitored in such experiments, as the samples are washed with a high concentration of salt, which displaces most CMG partner proteins (CMG itself is resistant to the high salt wash).

“3. In Figure 6B, there may be a mistake in the time courses, 10' release/20'/30'/40'/50'/40'/80'/120' ---. The 2nd 40' will be correctly 60'.”

Thank you for pointing this out – the switch in timings was to show the time after re-addition of alpha factor (to stop cells entering S-phase of the second cell cycle), but we have now simplified the figure (now Fig 7B) so that all times refer to release from the first G1-arrest.

“4. Is the Rrm3/Pif1-mediated pathway for old CMG dependent on Top2 activity?”

We would predict that Top2 is not required for the Rrm3/Pif1 pathway of old CMG disassembly, since Top2 plays a redundant role with Top1 in supporting replication fork progression. We have not tested directly whether Top2 is required for old CMG disassembly and feel that doing so would be beyond the scope of our revision.

“5. Does the Rad52 foci in mcm7-10R increase or decrease in the absence of Rrm3?”

The proportion of cells with Rad52 foci in mcm7-10R does increase in combination with rrm3 Δ , to a similar level to dia2 Δ . We now include these new data in Figure 6C of the revised manuscript.

- Any additional non-essential suggestions for improving the study (which will be at the author's/editor's discretion)

“1. If possible, it might be better to shorten the description of identifying the ubiquitylation domain of Mcm7 (Fig. EV2&3) in the first section of 'Results' part.”

We have now reworked the description of the mapping experiments, following additional suggestions from Referee 3 and Referee 1. We have moved the key data to a new Figure 1 and included a cartoon that summarises the main findings (Fig 1F).

“2. At the line 235, it might be better to clearly state the names of 'other replisome proteins' (Ctf4, Mrc1, Pol ϵ) not only in the figure legend but also in the text.”

We now do so in the revised text on line 250.

“3. At the line 363-364, it was stated that "When dia2 Δ cells progress through S-phase, old CMG complexes from the previous cell cycle are removed by an unknown pathway." Since this is just a speculation at that point, it is better to add "(Fig EV1B).”

We now cite Fig EV1B in the rewritten version of this sentence (lines 382-384).

“4. Is it possible to remove old CMG in dia2 Δ cells by exogenously expressed Dia2 in G1-phase?”

Yes! We showed previously (Figure 6B of Maric et al, 2014, *Science*; compare samples 2 and 3) that re-expression of Dia2 during G1-phase in dia2 Δ cells induces rapid and efficient disassembly of old CMG complexes.

This provides functional proof that old CMG complexes are fully competent for ubiquitylation by SCF-Dia2 and disassembly by Cdc48, even during G1 phase. SCF-Dia2 acts specifically on old CMG but is not regulated by the cell cycle.

We now refer to these data on lines 110-113.

Referee #3:

In this manuscript, Rivera and colleagues report the characterization of a fundamental mechanism by which eukaryotes unload the replicative helicase CMG from chromatin upon termination of DNA replication. Previous studies from the Labib lab and from others have shown that converging CMG helicases bypass each other at replication termination sites and remain topologically trapped around DNA, until they are ubiquitylated and displaced from chromatin by Cdc48. In budding yeast, the ubiquitylation of the CMG subunit MCM7 depends on the ubiquitin ligase SFCDia2. Failure to disassemble CMG complexes in the absence of Dia2 results in genomic instability and growth defects. Because SFCDia2 targets several other DNA replication and repair factors, the deleterious effect of the persistence of trapped CMG complexes on DNA has not been formally established.

Here, the authors used recombinant CMG and a reconstituted ubiquitylation assay to identify critical Mcm7 residues targeted by Dia2. With this novel information in hand, they derived a largely non-ubiquitylatable version of Mcm7 (Mcm7-10R), which was used for further in vitro and in vivo studies. They show that cells expressing this allele phenocopy *dia2* Δ mutants, especially when combined with the ablation of the Dia2 TPR domain, involved in its recruitment to the replisome. Interestingly, Mcm7-10R cells accumulated CMG on DNA after DNA replication but were viable due to a second pathway involving helicases of the Pif1 family. In particular, the authors show that the Rrm3 helicase is essential for viability in the absence of Mcm7 ubiquitylation, and that Rrm3 exerts this critical function during DNA replication by removing 'old' CMG complexes that remain on chromatin after the previous S phase.

Overall, the data presented in the manuscript are of high quality and support the view that two distinct pathways coexist in eukaryotic cells to degrade CMG helicases after completion of DNA replication. The first pathway is an ancestral pathway that relies on a conserved family of DNA helicases to remove old CMG complexes during DNA synthesis. The second pathway depends on less conserved enzymes that ubiquitylate CMG at the end of the S phase to promote its degradation by Cdc48/p97 and the proteasome. These findings represent a major advance in our understanding of this important cellular process. However, specific issues need to be addressed to strengthen their case.

We thank the referee for his / her interest in our work and for helpful comments below.

1. The first third of the manuscript describes the strategy used to identify the major ubiquitylation sites of Mcm7, which were then mutated to generate the Mcm7-10R construct. The strategy used to identify these sites is very elegant and it is a pity that at least some of these data are not presented in a main figure.

Thanks for this helpful suggestion – we now include a new Figure 1 based on data that were formerly in Fig EV3B-G. We also include a new cartoon in Fig 1F to summarise the key findings.

2. Two figures are listed as Appendix Figures (S1 and S2), but Appendix Figures S1 is not even mentioned. These figures should be listed as EV figures so as not to unnecessarily complicate the structure of the manuscript.

Actually, we did cite both supplementary figures in the manuscript (Appendix Fig S1 was cited on lines 215-216 and Appendix Fig S2 on line 315).

In our initial submission, we followed the EMBO J. guidelines that only allow up to five EV figures – this was why we had to include Appendix Figures S1-S2. However, we have now reorganised the figures in the revised version of our manuscript, to include Referee 3's useful suggestion of including the ubiquitylation mapping data in a main figure (the new Figure 1), which saves us some space in EV figures.

Therefore, we have moved what was Appendix Fig S1 to what is now Fig EV2C-D and have moved what was Appendix Fig S2 to what is now Fig EV4.

3. Line 257: should read (Fig 3A and 3D).

Thank you – we have corrected this on line 272 of the revised manuscript (it's now Fig 4A and Fig 4D).

4. Line 263: The authors should explain why Cdc48 is depleted in HU-arrested cells and not in late G1 (Fig 3A). Would this interfere with the G1/S transition?

There was no compelling reason for this - we simply wanted to get the cells as close as possible to the DNA replication termination step before we inactivated Cdc48.

The experiment would also have worked if we had inactivated Cdc48 in G1-phase.

5. In the experiment shown in Figure 3D, cells were synchronously released from an alpha-factor arrest into S phase. However, a large fraction of the cell population (~40%) appears to remain blocked in G1. This is surprising because these cells correctly degrade the CDK inhibitor Sic1. Although this does not affect the message of the figure, the authors should provide an experiment showing complete release from G1 arrest, as is the case in Figure 3A.

There was some minor issue with the growth medium when those experiments were performed, and we found it hard at that time to avoid the issue that the referee refers to. As noted by the referee, this did not affect the message of the figure. Therefore, we hope that the referee will understand that we would prefer not to delay publication of our manuscript by repeating this experiment.

6. Line 370: The authors should also cite a recent article from the Aharoni lab showing that Rrm3 unwinds G4 on the leading strand while Pif1 acts on the lagging strand (PMID: 38117984).

Thank you for pointing this out - we now cite this paper on lines 64-65.

Regarding strand specificity of action of Pif1 / Rrm3, we also note that the cited papers from the Coster lab (Williams) and Remus lab (Kumar) show via reconstituted in vitro reactions that Pif1 can also act on the leading strand template (the Remus paper shows that Pif1 can act on both strands). Moreover, the resolution of G4 structures takes place behind the replisome and thus is distinct from the displacement of barriers such as old CMG in front of the replisome, which is the subject of our manuscript.

7. In Fig EV5, the authors present different scenarios of CMG disassembly in different contexts, which are further discussed at the end of the Discussion section. It has recently been proposed by the Cortez lab that CMG is trapped on DNA by RAD51 during fork reversal, while the rest of the replisome disassembles. In such a configuration, would a converging fork disassemble this CMG in a manner compatible with proper termination of DNA replication?

The interesting model proposed by the Cortez lab in Fig S2D of Liu et al (2023, Science) suggests that CMG embraces the leading-stand template DNA as normal, at the front of the remodelled fork. If this were true, convergence with a leftward-moving fork (approaching from the right-hand side of the figure) would lead to CMG disassembly via CUL2-LRR1 and p97 as usual, since this pathway only requires unwinding of the intervening parental duplex.

Though very interesting, it's hard to see how this model relates to the disassembly of old CMG that is the focus of our manuscript, and we feel that this topic would be best discussed elsewhere.

Dear Prof. Labib,

Thank you for submitting a revised version of your manuscript. Your study has now been seen by all original referees, who find that their previous concerns have been addressed and now recommend publication of the manuscript. There remain only a few mainly editorial points that have to be addressed before I can extend formal acceptance of the manuscript:

- Please make sure to provide all the requested Source data files listed in the uploaded and attached Source Data checklist file, which you had been sent by my colleague Hannah Sonntag. Please complete the Source Data checklist and upload it to our online system. Source data files need to be saved in a scheme one figure/folder and then uploaded as .zip files. E.g. all the Source data files for figure 1 need to be saved in a single folder and this needs to be zipped and then uploaded as "SD figure 1.zip" file.

- Figure Legends: Please define the white arrows in the legend of figure 5e and provide the exact p values in the legend of figure 5f.

With best regards,

Cornelius

Cornelius Schneider, PhD
Editor
The EMBO Journal
c.schneider@embojournal.org

We realize that it is difficult to revise to a specific deadline. In the interest of protecting the conceptual advance provided by the work, we recommend a revision within 3 months (5th Sep 2024). Please discuss the revision progress ahead of this time with the editor if you require more time to complete the revisions. Use the link below to submit your revision:

Referee #1:

I felt that the original manuscript was already incredibly strong. The authors have done an excellent job of addressing my concerns. I have no remaining concerns.

Referee #2:

Now I am convinced by the authors' explanations and responses for all the issues that I had raised, and I think the revised version is suitable for publication in the EMBO Journal.

Referee #3:

The authors have adequately addressed the issues I raised and in my opinion the manuscript is now suitable for publication in the EMBO Journal.

All editorial and formatting issues were resolved by the authors.

Dear Prof. Labib,

I am pleased to inform you that your manuscript has been accepted for publication in the EMBO Journal.

Yours sincerely,

Cornelius

Cornelius Schneider, PhD
Editor
The EMBO Journal
c.schneider@embojournal.org
